# LARGE1 processively polymerizes length-controlled matriglycan on prodystroglycan

Soumya Joseph [1,2,3], Nicholas J. Schnicker [2,4], Nicholas Spellmon [5], Zhen Xu [4], Rui Yan [5], Zhiheng Yu [5], Omar Davulcu [6], Tiandi Yang [1,2,3], Jesse Hopkins [7], Mary E. Anderson [1,2,3], David Venzke [1,2,3] & Kevin P. Campbell [1,2,3] ✉

Matriglycan is a linear glycan (xylose-$\beta$1,3-glucuronate)$_n$, which binds proteins in the extracellular matrix that contain laminin-globular domains and Lassa Fever Virus. It is indispensable for neuromuscular function. Matriglycan of insufficient length can cause muscular dystrophy with abnormal brain and eye development. LARGE1 (Like-acetylglucosaminyltransferase-1) uniquely synthesizes matriglycan on dystroglycan. The mechanism of matriglycan synthesis is not obvious from cryo-EM reconstructions of LARGE1. However, by reconstituting activity in vitro on recombinant prodystroglycan we show that the presence of the dystroglycan N-terminal domain (DGN), phosphorylated core M3, and a xylose-glucuronate primer are necessary for matriglycan polymerization by LARGE1. By introducing active site mutations, we demonstrate that LARGE1 processively polymerizes matriglycan on prodystroglycan, with its length regulated by the dystroglycan prodomain, DGN. Our enzymatic analysis of LARGE1 uncovers the mechanism of matriglycan synthesis on dystroglycan, which can form the basis for therapeutic strategies to treat matriglycan-deficient neuromuscular disorders and arenaviral infections.

Matriglycan is a linear polysaccharide composed of alternating xylose and glucuronate[1] that is polymerized by LARGE1[2] and LARGE2[3] glycosyltransferases post-translationally on the heavily and heterogeneously O-glycosylated[4–6] mucin-like domain of dystroglycan. Dystroglycan is a component of the dystrophin-glycoprotein complex[7,8] that connects the cytoskeleton to the extracellular matrix (ECM)[9,10] by virtue of matriglycan, which binds ECM proteins that contain laminin-globular (LG) domains[11], such as laminin[12], agrin[13,14], perlecan[15,16], neurexin[17,18], pikachurin[19,20], EYS[21] and SLIT2[22] as well as the Lassa Fever spike glycoprotein[23–25]. Matriglycan on dystroglycan is indispensable for neuromuscular function, brain development[26–28], and efficient arenaviral entry[29]. Conversely, matriglycan of insufficient

length causes muscular dystrophies that are sometimes accompanied by intellectual disability[30,31]. Currently, there are no treatments for matriglycan-insufficient muscular dystrophies or Lassa Fever viral infections[32–35].

Gene therapies may offer the most effective solutions for genetic disorders[36]. However, LARGE1 gene transduction assays produce matriglycan of indiscrete lengths—a continuous smear that does not resemble physiologically observed sizes, inferred by discrete bands with a clear migration front and rear, on Western blots[37]. In contrast to many natural biopolymers—polynucleotides and polypeptides—whose length is typically governed by a template, LARGE glycosyltransferases polymerize matriglycan of discrete lengths without a corresponding

[1]Senator Paul D. Wellstone Muscular Dystrophy Specialized Research Center, University of Iowa Roy J. and Lucille A. Carver College of Medicine, Iowa City, IA, USA. [2]Department of Molecular Physiology and Biophysics, University of Iowa Roy J. and Lucille A. Carver College of Medicine, Iowa City, IA, USA. [3]Department of Neurology, University of Iowa Roy J. and Lucille A. Carver College of Medicine, Iowa City, IA, USA. [4]Protein and Crystallography Facility, University of Iowa, Iowa City, IA, USA. [5]Janelia Research Campus, Howard Hughes Medical Institute, Ashburn, VA, USA. [6]Pacific Northwest National Laboratory, Environmental Molecular Sciences Laboratory, Richland, WA, USA. [7]BioCAT Beamline 18-ID-D, Advanced Photon Source, Argonne National Laboratory, Lemont, IL, USA. ✉e-mail: kevin-campbell@uiowa.edu

polymeric template[37]. Bifunctional glycosyltransferases with tandem catalytic domains that form dimers, such as exostosins (EXT1-EXT2 and EXTL3$_2$)[38–41] and chondroitin polymerase (K4CP)[42], appear to synthesize their respective linear glycans distributively (enzyme dissociates from substrate between catalytic cycles) on synthetic substrates. In this study, we sought to determine the mechanism of LARGE1 matriglycan polymerization and length control. Using single particle cryo-EM reconstructions, we and others[43] show that LARGE1 has a similar architecture to these bifunctional glycosyltransferases, in which the active sites on each protomer face opposite directions. However, the mechanism of matriglycan synthesis and length control is not obvious from cryo-EM reconstructions of LARGE1.

In this work, we investigated the mechanism by which LARGE1 polymerizes matriglycan on dystroglycan in vitro using soluble recombinant purified LARGE1dTM[2] and an engineered substrate, prodystroglycan, in terms of processivity versus distributivity[44] and the biochemical factors that regulate polymer synthesis and length. We first confirm the structural integrity of our LARGE1 construct by light scattering techniques and single particle cryo-EM reconstructions. We subsequently show that LARGE1dTM only synthesizes discrete sizes of matriglycan on prodystroglycan if (1) the mucin-like domain is contiguous with the dystroglycan N-terminal domain (DGN)[45], (2) core M3 is phosphorylated on carbon-6 [N-acetylgalactosamine–β3-N-acetylglucosamine–β4-(phosphate-6-)mannose][30] and (3) a xylose-glucuronate primer is present[46,47]. Additionally, using a mixture of LARGE1 active site mutants, we show that LARGE1 polymerizes matriglycan processively on prodystroglycan but distributively on a synthetic substrate. And, using a neuromuscular patient mutation (T192M)[31], we show that matriglycan length is controlled by the dystroglycan prodomain, DGN. Collectively, our study specifies the mechanism and factors that control matriglycan polymerization by LARGE1 on dystroglycan, which will underpin therapeutic strategies to treat neuromuscular disorders and arenaviral infections by modulating matriglycan length.

## Results

### In vitro LARGE1dTM matriglycan synthesis assay on recombinant prodystroglycan

We have historically relied on synthetic small-molecule substrates, such as 4-methylumbelliferone (4MU) or biotin-conjugated sugars, to assay LARGE1 glycosyltransferase activity because the products are conveniently separated from the starting material by chromatographic techniques and easily detected. However, such substrates are unlikely to accurately recapitulate the characteristics of glycosyltransferases on their cognate endogenous targets. For example, the range of matriglycan lengths polymerized on 4MU-glucuronate-xylose by LARGE1dTM follows a Poisson distribution[2] (Fig. 1a and Supplementary Fig. 1) and does not resemble the well-defined polymer lengths observed in vivo[26], which are suggested by the discretely migrating band of native dystroglycan modified with matriglycan on Western blots. Because there is a paucity of appropriate substrates that reflect the configuration of dystroglycan found in the Golgi, where it would naturally encounter LARGE1 (Fig. 1b), we decided to engineer a native-like, recombinant, soluble construct, prodystroglycan. To mimic the immature, but relevant configuration of dystroglycan found in the late-ER/early Golgi stages, we mutated the furin cleavage site (R311A/R312A)[48,49], so that dystroglycan retains the prodomain, DGN. The construct terminates prior to the transmembrane region. It preserves the autoproteolytic activity of the sea-urchin sperm protein, enterokinase and agrin (SEA) domain (Fig. 1c), which cleaves dystroglycan into alpha and beta chains[50]. Because alpha and beta dystroglycan remain tightly associated after cleavage, we used the engineered C-terminal hexahistidine-tag for nickel affinity purification (Fig. 1d). The recombinant protein is post-translationally modified by appropriate enzymes, including LARGE1, that are endogenous to HEK 293 Freestyle cells as it traverses the secretory pathway (Fig. 1b–d). We used batch anion exchange chromatography to recover a fraction of prodystroglycan devoid of matriglycan, synthesized by the host, for subsequent enzymatic assays (Fig. 1d). We validate this construct by showing that purified recombinant LARGE1dTM can polymerize matriglycan of defined length on prodystroglycan in vitro when provided with both UDP-xylose and UDP-glucuronic acid (Fig. 1e).

### Structural characterization of LARGE1dTM$_2$ used in glycosyltransferase assays

LARGE proteins consist of independent xylose and glucuronic acid transferase domains arranged in tandem (Fig. 2a). We confirmed the structural integrity of the LARGE1dTM construct[2] used in enzymatic assays through light scattering experiments and single-particle cryo-EM reconstructions (Fig. 2). The tandem Rossmann-like fold of the catalytic domains of LARGE1 (Supplementary Fig. 2a) was built into the reconstructed volumes (Fig. 2b). We found that LARGE1dTM forms a dimer in which the active sites on each protomer face opposite directions (Fig. 2c)[43]. Our reconstructed volume of apo-LARGE1dTM does not differ globally from that of LARGE1dTM-DAG1$_{28-340}$ enzyme-substrate (ES) complex, except for a density for UDP-glucuronic acid (Fig. 2d and Supplementary Fig. 4) which only binds the latter (Fig. 2e). Despite collecting data at zero- and thirty-degree tilt for the ES complex, one of the xylose transferase domains in both apo-LARGE1 and the ES complex remained poorly resolved without imposed symmetry (Fig. 2f). Interestingly, the residues in LARGE1 which when mutated have the most effect on the matriglycan synthesis map to the region of the xylose transferase domain that remains unresolved[51]. Although we reconstructed the volumes in both C1 and C2 symmetries, we suspect that the asymmetry of the LARGE1 homodimer is biologically relevant[52,53] and may hint at the stoichiometry of the LARGE1-prodystroglycan ES complex but we never observed nor could isolate the structurally symmetric dimer for comparison to determine its function. The density for prodystroglycan (DAG1$_{28-340}$), however, remains unresolved in the reconstruction, likely due to its dynamic nature[49], but its presence is implied by density for UDP-glucuronate in both glucuronic acid transferase active sites (Fig. 2e). Additionally, the N-glycans (Supplementary Fig. 3), coiled-coil domain, and N-terminal tail in both reconstructions also remain unresolved, likely because they were averaged to the level of noise during particle alignment due to their dynamics relative to the globular catalytic head domains.

We confirmed that LARGE1dTM dimerized in solution independently of its coiled-coil domain (Supplementary Fig. 2b) and N-glycans, by comparing it to LARGE2dTM, which lacks the coiled-coil domain (Supplementary Fig. 4), and digestion with PNGase F, respectively, using on-line multi-angle light scattering (SEC-MALS; Supplementary Fig. 2b). LARGE2dTM was not analyzed further in this study because deletion of LARGE1, but not LARGE2, causes neuromuscular pathology. We used small-angle X-ray scattering (SAXS) to generate a low-resolution molecular envelope of apo-LARGE1dTM (Supplementary Figs. 2c, 6-11 and Supplementary Tables 1, 3, 4) to show a stem domain attached to a globular head (Supplementary Fig. 2c), which we conclude represents the coiled-coil domain attached to the catalytic domains and is consistent with the expected topology of LARGE1 in the Golgi membrane (Fig. 2c). Conserved surface residues are primarily localized to the dimer interface (Supplementary Fig. 2d). The xylose and glucuronate transferase active sites coordinate a cation in apo-LARGE1dTM (Supplementary Fig. 2e). LARGE1dTM coordinates and maps were deposited with the following PDB and EMDB accession codes: 7UI6, EMD-26540 (C1 symmetry) and 7UI7, EMD-26541 (C2 symmetry). LARGE1dTM-DAG1$_{28-340}$ coordinates and maps were deposited with PDB ID 9E1T and EMD-47420.

We expect highly active enzymes from the structural integrity observed in both reconstructions of recombinant LARGE1dTM. However, despite elucidating the relative positions of the active sites in the reconstructed volumes, which imply that matriglycan is likely

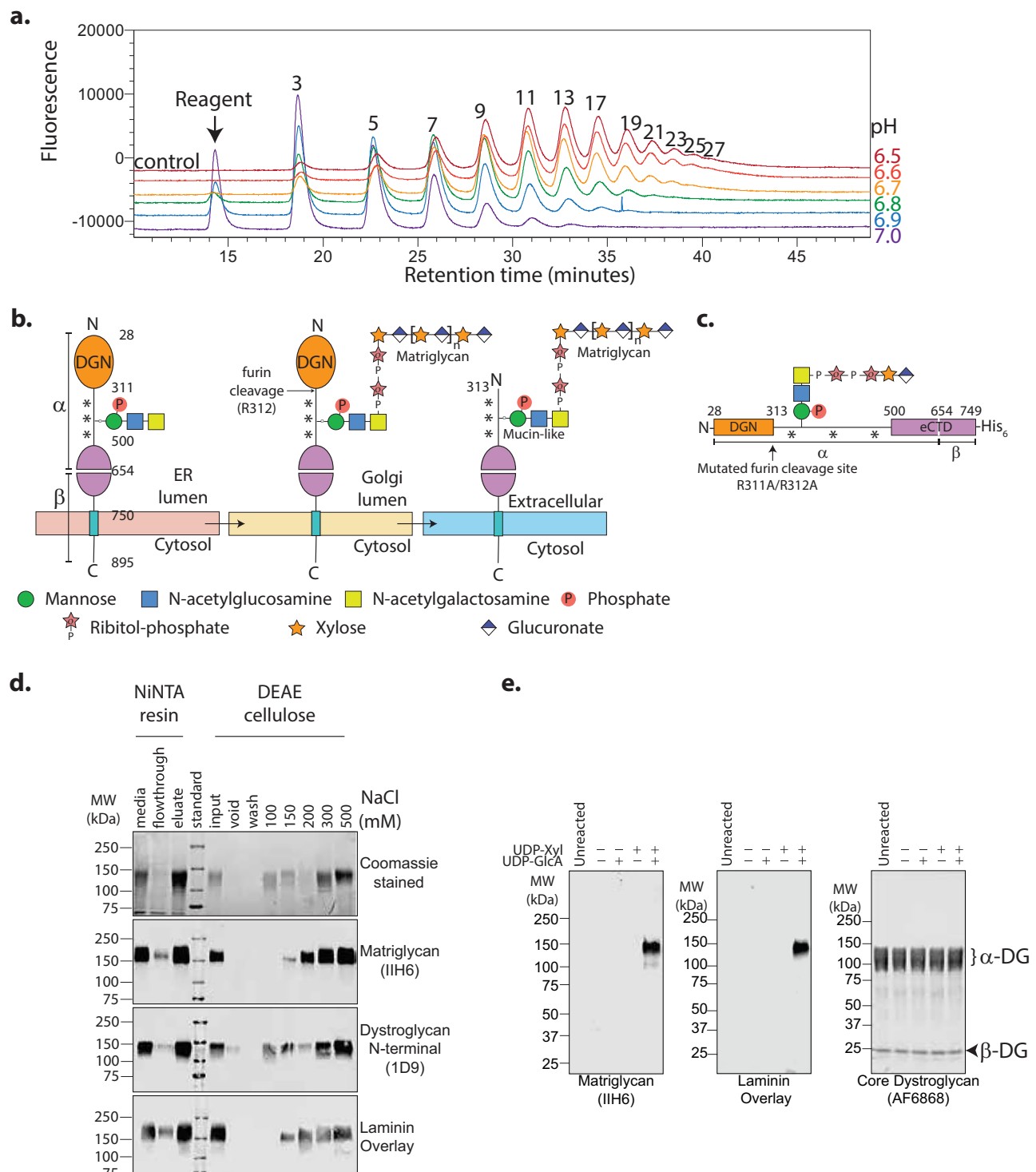

**b.** Legend:
- ● Mannose
- ■ N-acetylglucosamine
- ■ N-acetylgalactosamine
- ● P Phosphate
- ⬟P Ribitol-phosphate
- ★ Xylose
- ◆ Glucuronate

polymerized by switching between the orthogonal active sites on different protomers (Fig. 2c, arrow), we were unable to deduce whether matriglycan is synthesized processively or distributively on dystroglycan and how polymer length is controlled from the LARGE1 cryo-EM reconstructions.

**The mucin-like domain of dystroglycan must be contiguous with DGN and additionally carry a xylose-glucuronate (Xyl-β1,4-GlcA) primer for matriglycan synthesis by LARGE1**

To determine the biochemical factors necessary for matriglycan synthesis by LARGE1 on dystroglycan we assayed in vitro matriglycan

synthesis activity of recombinant LARGE1dTM on constructs of dystroglycan, (Fig. 3). To test whether LARGE1dTM can polymerize matriglycan on mature dystroglycan, in which DGN is absent, or whether DGN must remain contiguous with the mucin-like domain of dystroglycan, we expressed and purified recombinant constructs of wild-type (rDystroglycan, ΔDGN$_{313-749}$) or dystroglycan with a furin site mutation (rProdystroglycan, +DGN$_{28-749}$) for in vitro matriglycan synthesis assays (Fig. 3a). LARGE1dTM can polymerize matriglycan on prodystroglycan but not on mature dystroglycan (Fig. 3b), suggesting that the mucin-like domain of dystroglycan must be contiguous with DGN (Fig. 3a, b) for matriglycan synthesis by LARGE1.

**Fig. 1 | Engineered recombinant prodystroglycan is a native-like substrate for LARGE1 matriglycan synthesis in vitro. a** Anion exchange chromatogram of matriglycan polymerized by LARGE1dTM on 4-methylumbelliferyl-glucuronate-xylose in vitro. Product length (enumerated peaks represent the number of monosaccharides) is inversely proportional to the pH of the reaction and product abundance follows a Poisson distribution (Supplementary Fig. 1), which is characteristic of distributive polymerization. Similar experimental results were replicated at least three times independently. Source data are provided as a Source Data file. **b** Schematic of the relevant core M3 glycan modifications on dystroglycan acquired in the secretory pathway. Panels from left to right: the signal peptide (residues 1-27), which is cleaved in the ER, is omitted. Dystroglycan globular N-terminal domain (DGN; orange) and the autoproteolytic extracellular C-terminal domain (eCTD; purple) flank the heavily and heterogeneously O-glycosylated (***) mucin-like domain (black line). Threonine residues 317, 319, and 379 in human dystroglycan are post-translationally modified sequentially in the ER by POMT1/2, POMGNT2, B3GALNT2, and POMK to make phosphorylated core M3 (N-acetylgalactosamine-β3-N-acetylglucosamine-β4-(phosphate-6-) mannose). In the

Golgi, FKTN (FCMD) and FKRP extend core M3 by two phosphoribitol units followed by RXYLT1 (TMEM5) and B4GAT1 which add a xylose-glucuronate primer. DGN is cleaved in mature dystroglycan present on the cell membrane. O-glycans and a secondary matriglycan site on the mucin-like domain are omitted for clarity **c** A schematic of recombinant human prodystroglycan (DAG1$_{28-749}$), which is designed to retain DGN due to a mutation in the furin cleavage site (R311A/R312A; arrow) and is receptive to modification by LARGE1. **d** Purification of prodystroglycan using Ni-NTA resin followed by batch anion exchange chromatography, in which recombinant prodystroglycan modified by matriglycan elutes at higher ionic strength, was monitored by SDS-PAGE with Coomassie stain and Western blots for matriglycan (IIH6), dystroglycan N-terminal domain (anti-DGN; 1D9) and laminin overlay. Similar experimental results were replicated at least five times independently. **e** Western blots of matriglycan (IIH6 or laminin overlay) polymerized on prodystroglycan by LARGE1dTM in vitro with the addition of substrates, UDP-xylose (UDP-Xyl) and UDP-glucuronate (UDP-GlcA). Core proteins, α- and β-dystroglycan are indicated by a bracket and arrowhead, respectively. Similar experimental results were replicated at least three times independently.

---

Our previous work showed that LARGE1dTM was able to transfer glucuronic acid to xylose-α-pNP[2]. To test whether LARGE1 requires a xylose-glucuronate primer or whether a terminal xylose, naturally appended by RXylT1 (TMEM5), is sufficient for matriglycan synthesis, we digested prodystroglycan with β-glucuronidase[54] (Fig. 3c). LARGE1dTM could not transfer glucuronic acid to a lone, terminal xylose on prodystroglycan—a function that is usually performed by B4GAT1—and could not polymerize matriglycan in the absence of a xylose-glucuronate primer on prodystroglycan (Fig. 3d) consistent with previous findings[46,47].

### Phosphorylated core M3 is required for LARGE1 to efficiently synthesize matriglycan on dystroglycan

Phosphorylation of mannose at carbon-6 on core M3 by Protein O-mannose Kinase (POMK) is critical for efficient matriglycan elongation by LARGE1[30,45] and subsequent laminin binding[55,56]. Mutations in POMK that decrease mannose phosphorylation in dystroglycanopathy patients produce short matriglycan[30,45]. We previously showed that LARGE1dTM binds phosphorylated core M3 (K$_D$ = 11.5 μM) better than its non-phosphorylated counterpart (K$_D$ > 90 μM)[30]. To test the effect of core M3 phosphorylation on LARGE1 matriglycan polymerization in vitro, we transduced wild-type and POMK KO HEK 293T cells with adenovirus encoding prodystroglycan (DAG1$_{28-749}$) and purified the secreted protein from the media for use in in vitro matriglycan synthesis assays. LARGE1dTM was unable to synthesize matriglycan efficiently on prodystroglycan purified from POMK KO cells, which lacks the phosphate modification on mannose-C6 of core M3, within the same timeframe as proprotein from wild-type cells (Fig. 3e). However, given longer reaction times (16 h), LARGE1dTM could polymerize matriglycan on prodystroglycan from POMK KO cells (Supplementary Fig. 12). The slow rate of matriglycan synthesis on prodystroglycan lacking mannose-6-phosphate suggests that phosphorylated core M3 accelerates matriglycan synthesis.

### Elucidating the mechanism of matriglycan synthesis using active site mutants of LARGE1

Processive polymerization refers to the consecutive additions to a chain without dissociation of the substrate and the enzyme between catalytic cycles—these phenomena are typically observed in polynucleotide (DNA/RNA polymerases), polypeptide (ribosome) or polysaccharide syntheses. LARGE1 is a bifunctional glycosyltransferase with separate xylose and glucuronate transferase domains[2]. To determine whether LARGE1 polymerizes matriglycan distributively or processively on dystroglycan, we exploited the independent glycosyltransferase activities of LARGE1 in both cell-based and in vitro assays (Fig. 4). Mutating the active sites in isolation results in constructs that either lack xylose (DXD1; D242N/D244N) or glucuronate

(DXD3; D563N/D565N) transferase activity but retains the activity of the auxiliary domain (Fig. 4a).

We used a commercially available Hap1 LARGE1 KO cell line and its wild-type counterpart to evaluate the matriglycan synthesis activity of a combination of LARGE1 active site mutants. We found that adenoviral transduction of Hap1 cells with wild-type LARGE1 resulted in matriglycan polymers of dispersed lengths on endogenous dystroglycan and did not recapitulate a discretely migrating band observed in the Hap1 parental control cell line, C631 (Fig. 4b). This suggests that overexpressing LARGE1 overrides the control of matriglycan length observed in wild-type cells consistent with LARGE1 gene therapy studies[36,37]. We nonetheless evaluated the effects of combining LARGE1 mutants that lacked either xylose transferase activity (DXD1) or glucuronate transferase activity (DXD3A/B) in LARGE1 KO Hap1 cells. We show that co-transduction with a mix of LARGE1 mutants could restore matriglycan synthesis on endogenously expressed dystroglycan (Fig. 4b). The reason we used single mutants DXD3A (D563N) and DXD3B (D565N) for adenoviral expression is that two independent vendors found the production of LARGE1 encoding D563N/D565N resulted in empty virions. Co-transduction of LARGE1 mutants produced combinations of DXD1$_2$ and DXD3[A/B]$_2$ homodimers as well as DXD1:DXD3[A/B] heterodimers, which were confounding to interpret. However, we include these data because of two interesting phenomena: (1) Transduction of LARGE1 KO Hap1 cells with the glucuronate transferase inactive mutant, DXD3B (D565N), results in a faint matriglycan-positive band, which may represent the addition of a single xylose to the xylose-glucuronate primer (Fig. 4b, pink arrow). This band is less pronounced when cells are transduced with DXD3A. And (2) Transduction of wild-type Hap1 cells with adenovirus encoding the xylosyltransferase-inactivated mutant (DXD1), but not DXD3 mutants, did not produce a high molecular weight smear and maintained a discretely migrating matriglycan-positive band of reduced size (Fig. 4b, green arrow). Although we do not have a mechanistic explanation for why the singly transduced DXD1 variant decreases matriglycan length, it suggests that the glucuronate transferase domain partly controls the length of matriglycan.

Because we could not conclude whether LARGE1 polymerized matriglycan distributively or processively from adenoviral co-transduction experiments in cells, we compared matriglycan synthesized in vitro by a mixture of mutants that lacked either xylose or glucuronate transferase activity to wild-type LARGE1dTM using 4-methylumbelliferone-glucuronate-xylose (MU-GX) and prodystroglycan as substrates. Unlike heterodimers of mutant LARGE1 that might be formed in co-transduction assays, we noted that recombinant homodimers (DXD1$_2$ and DXD3$_2$) were unlikely to exchange to form DXD1:DXD3 heterodimers in solution because LARGE1dTM dimers are

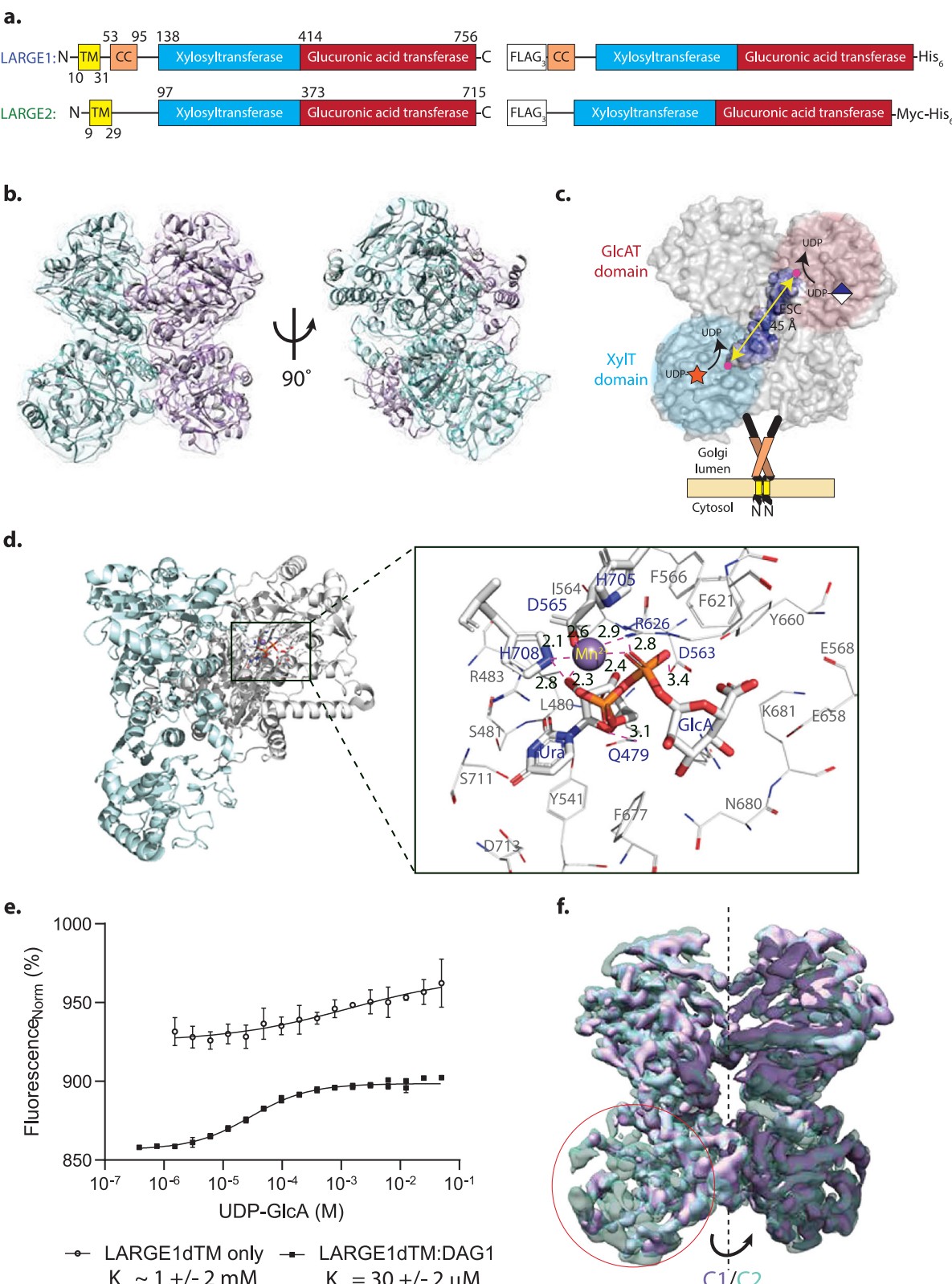

SDS-resistant in the absence of reducing agent (Supplementary Fig. 13). Additionally, using co-expression and subsequent tandem affinity purification of FLAG-tagged DXD1 and his-tagged DXD3, we show that heterodimers of DXD1:DXD3 and pseudomonomeric LARGE1dTM heterodimers, in which the active sites on one protomer is mutated, cannot polymerize matriglycan (Supplementary Fig. 14 and Fig. 4d).

Both wild-type LARGE1dTM and a mixture of active site mutants could polymerize matriglycan on 4-methylumbelliferone-glucuronate-xylose (Fig. 4c) and prodystroglycan (Fig. 4d). We found that the wild-type enzyme produced a discretely migrating band of matriglycan-modified prodystroglycan, as observed in vivo, in contrast to relaxed length control over short periods of time and high molecular weight smear given ample time observed for a mixture of LARGE1dTM active

**Fig. 2 | Reconstructed volume of LARGE1dTM glycosyltransferase. a** LARGE protein constructs, LARGE1dTM and LARGE2dTM (right panel), used for structural and biochemical assays carry an N-terminal FLAGx3 tag, C-terminal hexahistidine tag, and lack the transmembrane domain. **b** Two molecules of LARGE1 (grey cartoon) were refined into the 3.7-Å map of LARGE1dTM; protomers are displayed as cyan and purple transparent surfaces. **c** Orthogonal active sites each coordinating a manganese ion (magenta spheres) on alternate protomers face the same direction. A groove (highlighted surface) connects the xylose (cyan shading) and glucuronate (pink shading) transferase domains. The quaternary structure implies that matriglycan is polymerized using orthogonal active sites on alternate protomers of LARGE1 (yellow double-headed arrow). **d** Magnified view of a glucuronic acid transferase active site showing residues (sticks, labelled in blue text) that coordinate manganese (violet sphere) and UDP-glucuronate (sticks labelled "Ura" and "GlcA") in the LARGE1dTM-prodystroglycan (DAG1$_{28-340}$) reconstruction. Coordination distances (magenta dotted lines, Å) are in black text. Other residues that compose the active site are shown as lines with grey labels. **e** Titration of UDP-GlcA into NHS-red labeled LARGE1dTM alone (open circles) or saturated with prodystroglycan (DAG1$_{28-749}$, closed squares) observed by microscale thermophoresis. Data points and error bars represent the averages and standard error for $n = 3$ replicates, respectively. Similar experimental results were obtained independently four times. Source data are provided as a Source Data file. **f** Partial absence of density in the xylose transferase domain (red circle) when reconstructed with no symmetry imposed (purple, C1) aligned with a volume refined with axial symmetry (transparent cyan, C2).

site mutants (DXD1$_2$ and DXD3$_2$; Fig. 4d), which are forced to polymerize matriglycan distributively. This suggests that LARGE1 polymerizes matriglycan processively on its native-like substrate, prodystroglycan, but acts distributively on synthetic substrates. Moreover, in the context of the LARGE1-prodystroglycan ES complex, length control is linked to processive matriglycan polymerization.

### An enzyme-substrate complex between LARGE1 and DGN controls matriglycan length

To investigate how LARGE1 controls matriglycan length synthesized on prodystroglycan, we leveraged a patient mutation in DGN[T192M] which produces dystroglycan modified with short matriglycan compared to wild-type, that also migrates as a discrete band on Western blot[31]. The peptide backbone of the crystal structures for mouse DGN[T190M] aligns well with wild-type protein (Fig. 5a). The threonine to methionine mutation obstructs a cleft on DGN that would otherwise be of a greater length (Fig. 5b). Human recombinant prodystroglycan[T192M] carries short matriglycan (Fig. 5c, d), suggesting that matriglycan length control is intrinsic to DGN. Western blot analysis shows that although there are discrete matriglycan-positive bands from the analogous T190M mutant mouse tissues, the mutation results in matriglycan that migrates at correspondingly lower molecular weight compared to wild-type dystroglycan (Fig. 5e). Taken together, these results imply that DGN controls matriglycan length in the context of the LARGE1-prodystroglycan ES complex.

We were unable to polymerize matriglycan on prodystroglycan[T192M] in vitro under the same conditions used for the wild-type proprotein (Supplementary Fig. 15a-c), suggesting that other factors may aid matriglycan polymerization in vivo. To determine whether prodystroglycan[T192M] can form a competent ES complex with LARGE1, like wild-type proprotein, we showed that both wild-type and T192M prodystroglycan can bind LARGE1dTM with similar affinity (Supplementary Fig. 15d). We further showed that, whereas LARGE1dTM in isolation cannot bind UDP-glucuronate, the LARGE-prodystroglycan ES complex formed using either wild-type or a T192M variant bound UDP-glucuronic acid (Supplementary Fig. 15e). Because prodystroglycan[T192M] retains its backbone structure[57,58], binds to LARGE1 and forms an ES complex capable of binding UDP-glucuronate, and carries a small amount of matriglycan, our observations support the hypothesis that the primary cause of short matriglycan is the obstruction of the cleft on DGN by methionine sidechain, which leads to a shorter cleft and indicates its potential function as a ruler that measures matriglycan length (Fig. 5f and Supplementary Fig. 15).

## Discussion

LARGE1 glycosyltransferase polymerizes matriglycan, the physiologically functional and, in most tissues except brain[59], the terminal modification on dystroglycan that is required for connecting the cytoskeleton to proteins of the ECM. The length of matriglycan is a critical determinant of neuromuscular development and Lassa fever viral entry and egress[60,61]. Therefore, we dissected the mechanism of matriglycan synthesis by LARGE1 on dystroglycan by reconstituting the minimal, active LARGE1-prodystroglycan ES complex in vitro, which recapitulated the discrete size of matriglycan-modified dystroglycan observed in animal tissues. We found that LARGE1 integrates biochemical cues for efficient matriglycan synthesis, including the presence of the prodomain, DGN, a xylose-glucuronate primer, and phosphorylation of mannose carbon-6 on core M3 by POMK. These are characteristics of a few glycosaminoglycan synthases, for which a xylose is phosphorylated by Fam20b to efficiently polymerize heparan and chondroitin sulfate[62,63], both of which have only been studied using synthetic small molecular substrates rather than native-like proteinaceous substrates[38-41].

The relative orientations of the xylose and glucuronate transferase active sites, which resemble other bifunctional glycosyltransferases such as EXT1-EXT2 and EXTL3 dimers[38-41], suggest that matriglycan is polymerized by the repetitive addition of monosaccharides from alternating active sites on opposite protomers of the LARGE1 dimer. While some mechanisms of controlling polymer lengths rely on cyclisation[62], capping[59,63,64] or molecular timers, we show that matriglycan size is controlled by a putative ruler[65,66] on DGN. Although the mechanistic details remain to be elucidated, we produced short matriglycan by substituting threonine with methionine in recombinant prodystroglycan[T192M], which obstructs and shortens a cleft in the prodomain, DGN (Supplementary Fig. 18c). Whereas threonine residues within unstructured, mucin-like domains in dystroglycan have a high probability of O-glycosylation—between 0.97 and 0.99—the probability of T192 glycosylation, which resides in a wellfolded globular domains, is predicted to be 0.085 by NetOglycan4.0[67]. The distance between orthogonal active sites appears irrelevant for processive matriglycan synthesis[38,39,41], but rather depends on whether the substrate remains associated with LARGE1 over consecutive monosaccharide additions[65,68,69] presumably via LARGE1-DGN protein-protein interactions and binding to phosphorylated core M3[30]. Our data suggest that LARGE1 works distributively on synthetic substrates[2,40], but processively polymerizes matriglycan on the native-like prodystroglycan.

In summary, we show that in the context of a LARGE1-prodystroglycan ES complex, matriglycan is polymerized processively, in a length-controlled manner. This work improves our mechanistic understanding of LARGE1 matriglycan synthesis on dystroglycan to provide a basis for therapeutics which can increase matriglycan length to alleviate neuromuscular diseases or shorten its length to decrease viral load in acute *Arenaviral* infection.

## Limitations

Although LARGE1dTM can recapitulate the synthesis of length-controlled matriglycan on purified recombinant prodystroglycan in vitro, this system is not representative of its true configuration in the Golgi where the LARGE1 interacts with prodystroglycan in the context of dystrophin-glycoprotein complex and both entities are anchored to the membrane[5].

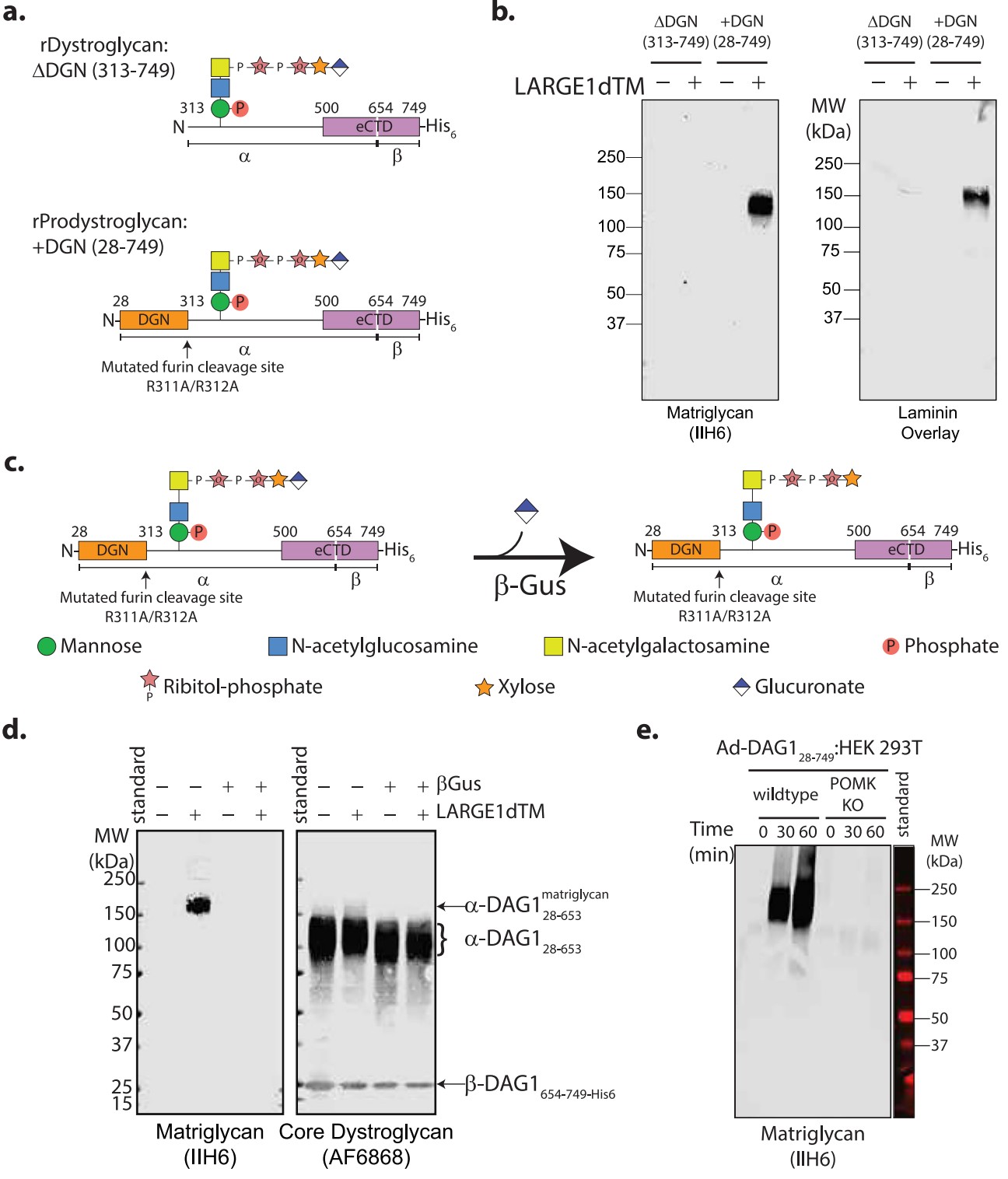

**Fig. 3 | Biochemical factors that promote matriglycan polymerization by LARGE1 on dystroglycan. a** Schematics of mature recombinant dystroglycan (rDystroglycan) in which DGN is cleaved (ΔDGN; DAG1$_{313-749}$) and recombinant prodystroglycan (rProdystroglycan) in which DGN remains covalently attached to the mucin-like domain through a peptide bond (+DGN; DAG1$_{28-749}$). **b** Western blot of matriglycan polymerized on mature dystroglycan (ΔDGN; DAG1$_{313-749}$) and prodystroglycan (+DGN; DAG1$_{28-749}$). **c** Schematic showing digestion of prodystroglycan with β-glucuronidase. Similar experimental results were obtained

independently at least three times. **d** Western blot of matriglycan synthesized in vitro by LARGE1dTM on prodystroglycan digested with β-glucuronidase (+) or control (-). Recombinant α- and β-dystroglycan fragments are marked. Similar experimental results were generated independently at least three times. **e** Western blot of matriglycan synthesized by LARGE1dTM in vitro on prodystroglycan. Wild-type or POMK KO HEK 293T cells were transfected with an adenovirus to express prodystroglycan (Ad-DAG1$_{28-749}$). Similar experimental results were obtained independently at least seven times.

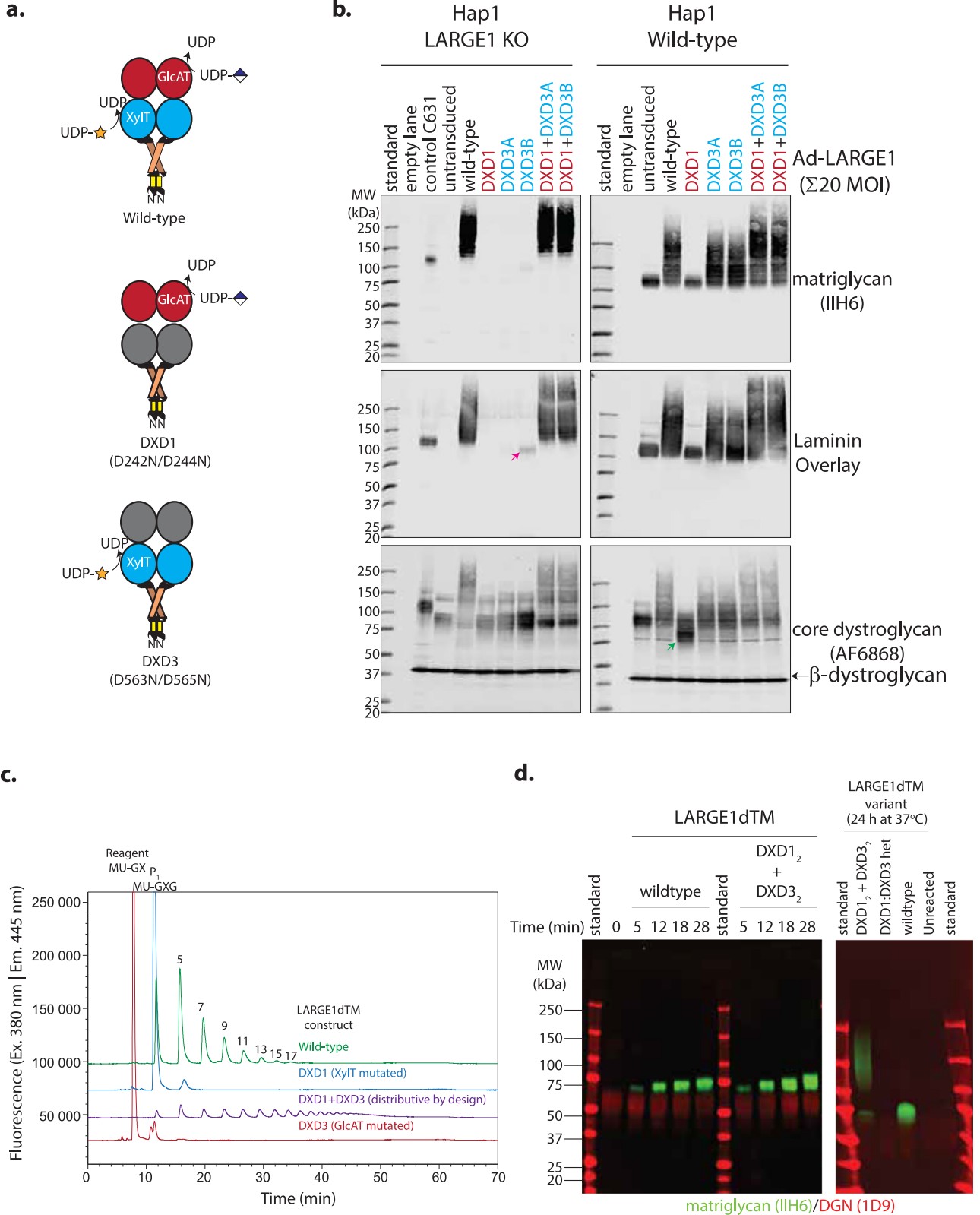

## Methods

### Ethical statement

Animal care, ethical usage, and procedures were performed in strict accordance with protocols approved by the National Institutes of Health and the Institutional Animal Care Use and Committee (IACUC), University of Iowa (#3051122). Mice were socially housed (unless single housing was required), under specific-pathogen-free conditions in an Association for Assessment and Accreditation of Laboratory Animal Care (AAALAC)-accredited animal facility. Mouse housing conditions were as specified in the Guide for the Care and Use of Laboratory Animals (National Research Council).

**Fig. 4 | Mixtures of LARGE1(dTM) active site mutants can polymerize matri-glycan. a** Schematic of wild-type and active site mutants of LARGE1 that either lack xylose transferase activity (DXD1) or glucuronate transferase activity (DXD3). **b** Western blot of *LARGE1* KO and wild-type Hap1 cell lines transduced with ade-novirus encoding LARGE1 (20 MOI): wild-type (WT), xylose transfer mutant (DXD1; D242N/D244N), glucuronate transfer mutant (DXD3A/B; D563N or D565N, respectively) or a combination of both (DXD1 + DXD3A/B). Matriglycan was detected using an anti-matriglycan antibody (IIH6) or laminin overlay; core dys-troglycan protein was detected using the AF6868 antibody. Pink arrow indicates a faint matriglycan-positive band in *LARGE1* KO Hap1 cells transduced with DXD3B only. Green arrow indicates a perturbation in the migration of dystroglycan in wild-type Hap1 cells transduced with DXD1. Similar experimental results were obtained independently at least three times. **c** Anion exchange chromatogram of matriglycan polymerized on 4-methylumbelliferone-glucuronate-xylose by LARGE1dTM or active site mutants DXD1, DXD3, or DXD1 + DXD3. Similar experimental results were obtained independently at least three times. Source data are provided as a Source Data file. **d** Western blot of matriglycan polymerized on prodystroglycan con-structs (as indicated) by a combination of LARGE1dTM active site mutants in the presence of UDP-sugars over time (left panel). Western blot of matriglycan poly-merized on prodystroglycan (DAG1$_{28-398}$) after 24 h by wild-type LARGE1dTM, DXD1:DXD3 heterodimers, and a mixture of DXD1$_2$ and DXD3$_2$ mutants. Similar experimental results were obtained independently at least seven times.

## Recombinant protein production

Pure recombinant LARGE proteins were obtained as previously described[2]. Briefly, LARGE1dTM and LARGE2dTM were secreted by stably transfected HEK293-F cells cultured in SFM II media supple-mented with glutamine and required antibiotics and were purified using TALON resin. To generate recombinant soluble constructs of dystroglycan, HEK293-F cells were stably transfected (X-tremeGENE™ 9 DNA Transfection Reagent; Sigma-Aldrich) according to the manu-facturer's protocol with pcDNA3.1+ encoding human dystroglycan residues 1–749, which included a mutation in the furin cleavage site (R311A/R312A) and a C-terminal hexahistidine-tag. The secreted dys-troglycan protein DAG1$_{28-749}$$^{R311A/R312A}$ was purified using His60 Ni IMAC resin (Takara) and subjected to anion exchange using DEAE resin to obtain a fraction of dystroglycan unmodified by matriglycan for enzymatic assays. Proteins were desalted into appropriate buffers for all other experiments. Proteins were snap-frozen in liquid nitrogen and stored at –80 °C.

## Antibodies

All antibodies used have been previously published: Anti-matriglycan (IIH6; 1:100 dilution) mouse IgM (Campbell laboratory/Developmental Studies Hybridoma Bank, Hybridoma product IIH6 C4, AB_2617216) RRID:AB_2617216. Anti-mouse IgM; goat polyclonal, Li-COR bios-ciences Cat# 926–32280, RRID:AB_2814919. Laminin mouse protein natural (ThermoFisher Scientific; 7.2 nM) Catalog #: 23017015. Anti-laminin antibody produced in rabbit (Millipore-Sigma L9393, RRI-D:AB_477163, 1:1000 dilution). Anti-dystroglycan N-terminal domain (1D9; 1:200 dilution) monoclonal mouse IgG anti-dystroglycan poly-clonal sheep IgG Catalogue #: AF6868 (R&D Systems; 1:200 dilution) RRID:AB_10891298. Anti-mouse IgG (H+L); donkey polyclonal, Li-COR Biosciences Cat# 926–32212, RRID:AB_621847. Anti-rabbit IgG (H+L); donkey polyclonal, Li-COR Biosciences Cat# 926–32213, RRI-D:AB_621848. All secondary antibodies were used at a dilution of 1:10,000.

## Cell lines

HEK 293 Freestyle cell line is from ThermoFisher Scientific, wildtype reference clone and LARGE1 KO Hap1 is from Haplogen/Horizon Dis-covery, wild-type and POMK KO HEK 293T is from Abcam (ab267313). Hap1 cells (RRID: CVCL_Y019) are a haploid human cell line with an adherent, fibroblast-like morphology, originally derived from parent cell line KBM-7 (RRID: CVCL_A426). Wild-type C631 (a diploid cell line containing duplicated chromosomes of Hap1) have been purchased from Haplogen/Horizon Discovery and gene-specific knockout Hap1 cells have been generated by Haplogen/Horizon Discovery. LARGE1 HAP1 cell KO lines are tested by rescue by transducing ade-novirally encoded WT LARGE1. Human POMK knockout HEK293T cell line (ab267313) was selected using puromycin. Mycoplasma testing of control (C631) on 9/24/2020 and LARGE1 KO on 9/24/2020 Hap1 cells as well as HEK 293 Freestyle parental line on 10/19/2020, and those expressing LARGE1dTM on 10/2/2020 and prodystroglycan on 10/2/2020 were performed to ensure the cells are not contaminated. Cell lines HEK 293T POMK KO and wildtype have been tested for myco-plasma by Abcam and are confirmed to be negative. The identity of Hap1 cells has been authenticated by the company using the STR profiling method.

## Size-exclusion chromatography with on-line multi-angle light scattering and small-angle X-ray scattering (SEC-MALS-SAXS)

SEC-MALS-SAX was performed at BioCAT beamline 18-ID at the Advanced Photon Source (Argonne National Laboratory, Lemont, IL). Proteins were centrifuged for 5 min at $18,000 \times g$. Samples (4–8 mg/mL, 300–500 µL) were applied to a 24 mL Superdex 200 Increase 10/300 GL column (Cytiva) in 20 mM HEPES pH 7.4 and 150 mM NaCl at a flow rate of 0.6 mL/min using a 1260 Infinity II HPLC (Agilent Technologies). In-line multi-angle light scattering (DAWN Helios II) with built-in dynamic light scattering and dif-ferential refractive index measurement (Optilab T-rEX) were used to determine weight-average molecular weight (Wyatt Technolo-gies) and degree of glycosylation (ASTRA 8). The sample passed through the UV detector (1260 Infinity II), MALS, and then the differential refractometer (Wyatt Technologies), in that order, before SAXS.

The SAXS flow cell consisted of a 1.0 mm ID quartz capillary with ~20 µm walls. A buffer sheath was co-flowed to separate the sample from the capillary walls and prevent radiation damage[70]. Scattering intensity was recorded using an Eiger2 XE 9M detector (Dectris) placed 3.67 m from the sample, which provided a q-range of 0.003–0.42 Å$^{-1}$. Half-a-second exposures were acquired every second.

Data were reduced using BioXTAS RAW 2.1.1[71] implementing ATSAS version 3.0.3. Exposures from protein peaks were buffer sub-tracted using the average of flanking regions. Peaks within the SEC-SAXS elution profile were deconvoluted into their components using evolving factor analysis with default settings as implemented in RAW[72]. The validity of the overall deconvolution was assessed on the com-ponent concentration profiles and mean error-weighted $\chi^2$ for the whole deconvolution range. Individual components were assessed based on the quality of the scattering profile and SAXS-derived molecular weights of the scattering profile compared to the expected values. The forward scattering intensity, I(0), and the radius of gyration ($R_g$) were calculated from the Guinier fit. The normalized Kratky plot, the pair-distance distribution plot $P$(r), and the corrected Porod volume were calculated using GNOM[73]. Low-resolution ab initio bead modeling was carried out for all samples using 15 reconstructions by DAMMIF[74] in slow mode, averaged by DAMAVER[75], and a final structure was refined in DAMMIN[76]. AMBIMETER was used to assess the ambi-guity of reconstructions[74]. The calculation of theoretical scattering curves for the model was performed by the program CRYSOL[77], which also determines the discrepancy ($\chi^2$ value) between the simulated and experimental scattering curves.

The LARGE1dTM, LARGE1dTM PNGase F-treated, LARGE2dTM, and LARGE2dTM PNGase F-treated SAXS data have been deposited in SASBDB under the accession codes SASDNF8, SASDNG8, SASDNH8 and SASDNJ8, respectively.

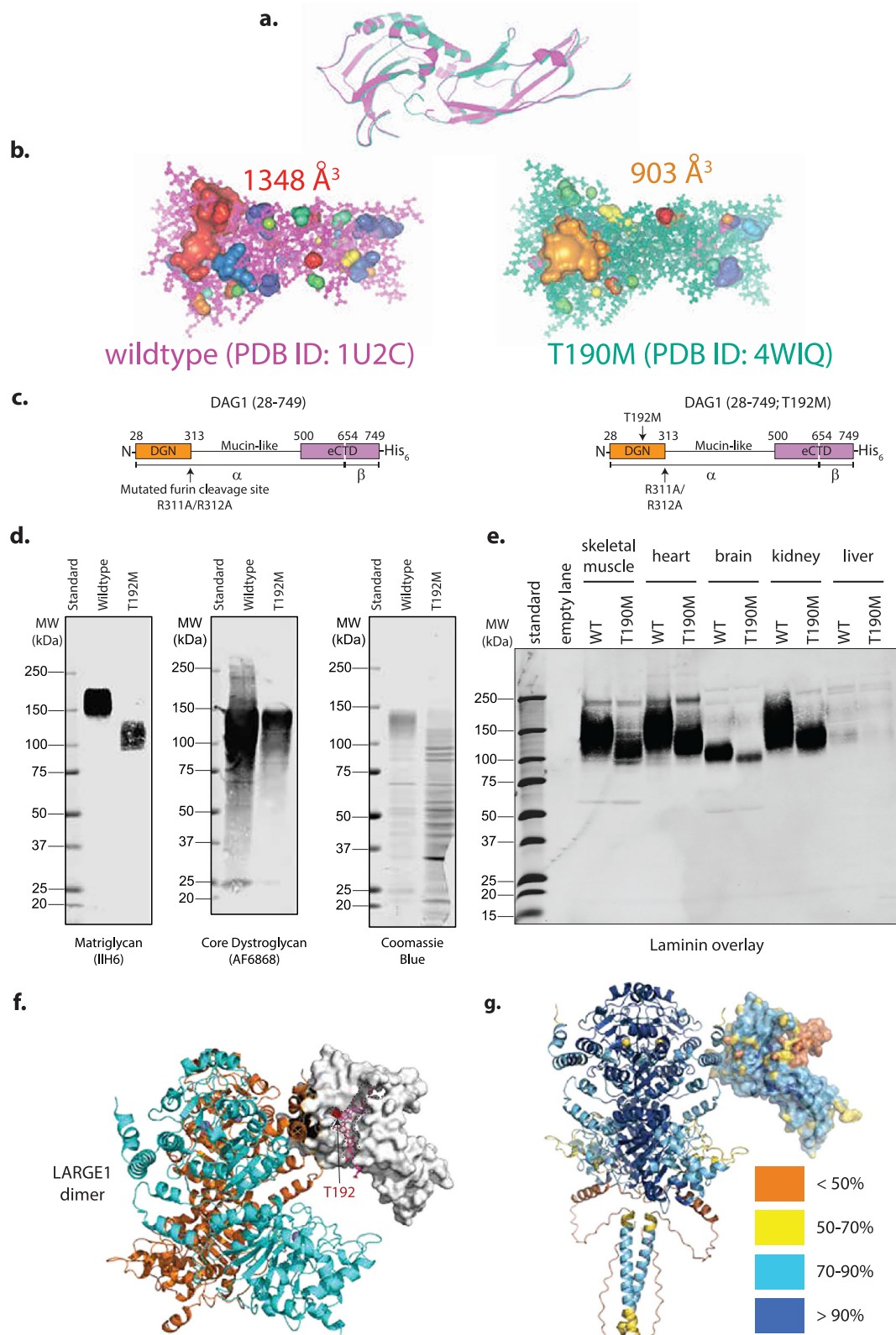

**a.**

**b.**

1348 Å³

903 Å³

wildtype (PDB ID: 1U2C)

T190M (PDB ID: 4WIQ)

**c.**

DAG1 (28-749)

28   313           Mucin-like   500   654  749
N—[DGN]——————————[eCTD]—His₆
      ↑              α                β
Mutated furin cleavage site
R311A/R312A

DAG1 (28-749; T192M)

T192M
↓
28   313           Mucin-like   500   654  749
N—[DGN]——————————[eCTD]—His₆
      ↑              α                β
R311A/
R312A

**d.**

Matriglycan (IIH6)

Core Dystroglycan (AF6868)

Coomassie Blue

**e.**

Laminin overlay

**f.**

LARGE1 dimer

T192

**g.**

< 50%
50-70%
70-90%
> 90%

## N-glycomics of recombinant LARGE1dTM

LARGE1dTM (200 μg) was digested by trypsin in a 50 mM ammonium bicarbonate buffer (pH = 8.4) overnight at 37 °C. N-glycans were released by PNGase F (New England Biolabs) and enriched by C18 solid-phase extraction. Briefly, a C18 Sep-Pak (Waters) cartridge was conditioned by successive washing of 5 mL methanol, 5 mL ultrapure water, 5 mL acetonitrile, and 15 mL ultrapure water. The reaction mixture was loaded onto the preconditioned cartridge, and the flow-through was collected and lyophilized. A 1.5 mL slurry of sodium hydroxide in DMSO and 0.6 mL of iodomethane were added to the lyophilized N-glycans. The mixture was vortexed at room temperature for 45 min before being extracted by chloroform. The chloroform solution was washed with ultrapure water three times and dried down under a stream of nitrogen. The permethylated N-glycan was dissolved in 50%

**Fig. 5 | LARGE1dTM recapitulates pathologically short matriglycan on prodystroglycan$^{T192M}$. a** Protein backbone of wild-type (magenta; PDB ID:1U2C) and T190M (teal; PDB ID: 4WIQ) mouse DGN are overlaid. **b** The volume of clefts in both crystal structures was calculated using Caver Analyst[86]. **c** Schematic of prodystroglycan representing human wild-type and T192M mutant constructs. **d** Western blots of wild-type and T192M recombinant prodystroglycan purified from HEK 293 Freestyle cells. Matriglycan was detected using an anti-matriglycan antibody (IIH6); core dystroglycan protein was detected using the AF6868 antibody. Results are representative of at least three independent experiments. **e** Laminin overlay of dystroglycan in the indicated tissues from wild-type and T190M mice. Results are representative of at least three independent experiments. **f** AlphaFold 3 model of DGN (white surface) bound to LARGE1 dimer (backbones of protomers are shown in orange and blue cartoon) with matriglycan (pink sticks) modeled into a cleft (dotted lines) that acts as a ruler to measure length. Threonine-192 is shown in red surface (black arrow). **g** Confidence of the AlphaFold 3 prediction is color-coded onto the cartoon backbone of the complex. Glycans have been removed from predicted models for clarity and are shown in Supplementary Fig. 16.

methanol and loaded onto another preconditioned C18 Sep-Pak cartridge. The cartridge was washed with 5 mL ultrapure water and 3 mL of 15% acetonitrile before eluting with 3 mL of 50% acetonitrile. The purified glycans were lyophilized and dissolved in 10 μL of methanol. One μL of the solution was mixed with 1 μL of 20 mg/mL 3,4-diaminobenzophenone and analyzed by a Bruker ultrafleXtreme MALDI-TOF/TOF.

### Glycoproteomics
LARGE1dTM (50 μg) was reduced by 100 μL of 50 mM of dithiothreitol and carbamidomethylated and by 100 μL of 100 mM iodoacetamide before dialyzed against 16 L (4 L each time, replaced buffer three times) of 50 mM ammonium bicarbonate with a 10 kDa cut-off dialysis cassette (Sigma-Aldrich). The solution was lyophilized and digested by 1 μg MS grade trypsin (Promega) in a 50 mM ammonium bicarbonate buffer (pH = 8.4) overnight at 37 °C. The buffer was lyophilized before the tryptic peptides were dissolved in 100 μL of 0.1% formic acid. Two μL of the solution was injected into EASY-nLC coupled to an orbitrap Fusion Lumos mass spectrometer. A two-solution (solution A: 0.1% formic acid; solution B: 80% acetonitrile, 0.1% formic acid) nanoLC gradient was used to elute an in-house packed C18 column: 3–7% B in 3 min, 7–20% B in 60 min, 20–42% B in 10 min, 42–60% B in 10 min, 80–98% B in 10 min. The solution B was kept at 98% until 100 min. The mass spectrometer was operated under the positive mode. The spray voltage was 1900 V. The ion transfer line temperature was 275 °C. MS resolution was set to 120 K and the mass range was 350–1500 Da. The maximum injection time was 100 ms. The AGC target was 200,000. S-lens RF level was 30. MS/MS resolution was 30 K, and the scan started at 80 Da. The maximum injection time was 90 ms. The AGC target was 80,000. The HCD collision energy was 34%. MS was active from 2 to 95 min. Glycopeptide spectra were manually analyzed.

### Cryo-EM sample preparation and data acquisition
LARGE1dTM, or LARGE1dTM mixed with equimolar prodystroglycan constructs and additives, were vitrified on UltrAuFoil R 2/2 200 mesh grids in 20 mM PIPES pH 6.6, 150 mM NaCl, 1 mM MgCl₂, 1 mM MnCl₂, and 1 mM CaCl₂ using a Vitrobot Mark IV (FEI). Grids were glow discharged for 60 s, −15 mA on a PELCO easiGlow (Ted Pella) system. Sample (3 μL) was applied to grids in the Vitrobot chamber (24 °C and 94% humidity) and blotted for 3 s before plunge-freezing in liquid ethane. Data were collected on a Titan Krios G3 microscope (300 kV) using SerialEM with a K3 direct electron detector (Gatan). A total of 5085 movies were collected at a pixel size of 0.40075 Å/pixel (super-resolution mode) with a dose of ~50 electrons/A², exposure time of 1.664 s, 50 frames, and a defocus range of −0.8 to −2.0 μm.

### Cryo-EM image processing and 3D reconstruction
Movies were subject to patch motion correction and patch CTF estimation in CryoSPARC[78]. Initial picks were performed on a subset of data using the blob picker followed by two-dimensional (2D) classification and ab initio reconstruction to generate three-dimensional (3D) templates for template-based picking on the full dataset. Particles (1,742,250) were extracted using a 300-pixel box size (0.8015 Å/pixel) and cleaned with multiple rounds of 2D classification. Multiple classes

were used for ab initio reconstruction followed by heterogeneous refinement to select 81,263 particles for non-uniform refinement[79] in C1 (Supplementary Fig. 16, EMD-26540). Refinement in C2 symmetry was also performed using 107,895 particles (Supplementary Fig. 16, EMD-26541). Processing was done initially in CryoSPARC version 3.1 and finished in version 3.3.1. Final maps were post-processed using DeepEMhancer[80] and used for model building and validation. LARGE1dTM in complex with DGN were processed in Relion 4.0 with particle coordinates input from CryoSPARC (Supplementary Fig. 17).

### Model building and refinement
An initial model for LARGE1 was generated by truncating regions from the LARGE1 model in the AlphaFold Protein Structure Database[81]. The model was initially docked into the density map using Fit in Map in Chimera[82]. Manual model building was performed in Coot[83] and refined using real-space refinement in Phenix[84] (2010). Both models (C1, PDB 7UI6 and C2, PDB 7UI7) were subject to comprehensive validation in Phenix. Figures were generated in Chimera and PyMOL. Software used for data processing, model building, and refinement except for CryoSPARC was curated by SBGrid[85].

### Adenoviral transduction
Wild-type and LARGE1 KO Hap1 cells were cultured in IMDM containing 2% FCS supplemented with glutamine and penicillin/streptomycin. Cells were transduced with adenovirus-5 (multiplicity of infection (MOI) of 20) encoding wild-type or mutant LARGE1 D242N/D244N (DXD1), D563N (DXD3A) or D565N (DXD3B). Cells were subsequently cultured in the same media containing 10% FCS for 48 h at 37 °C.

### Wheat-germ agglutinin (WGA) affinity chromatography
Cells were washed in PBS and then PBS-containing protease inhibitors (leupeptin, pepstatin A, aprotinin, and PMSF). Cells were scraped and incubated in 50 mM Tris pH 7.6, 150 mM NaCl 1% Triton X-100 with protease inhibitors. The solution was rotated for 1 h at 4 °C. The supernatant was applied to WGA resin and rotated for 10 min. The resin was washed three times with the same buffer containing 0.1% Triton X-100. The slurry was combined with SDS-loading dye and the sample was heated for 5 min at 99 °C.

### Matriglycan polymerization assays
Purified recombinant dystroglycan that was devoid of matriglycan or 4-methylumbelliferone-glucuronate-xylose (MU-GX; 0.4 mM) was combined with UDP-xylose (1 mM) and UDP-glucuronic acid (1 mM) with LARGE1dTM constructs (wild-type, DXD1 or DXD3) in 20 mM PIPES pH 6.6, 150 mM NaCl, 2 mM CaCl₂, 2 mM MgCl₂, 2 mM MnCl₂ and incubated for 3–16 h at 37 °C. The reaction in Fig. 5c was carried out in a buffer with 20 mM HEPES, pH 7.4, and 300 mM imidazole instead of PIPES. Reactions were terminated by heating to 99 °C or the addition of SDS-loading dye for substrates 4-methylumbelliferone or recombinant dystroglycan, respectively.

### Anion exchange chromatography
Samples with matriglycan polymerized on MU-GX were resolved on an anion exchange column (Phenomenex SphereClone™ 5 μm SAXS 80 Å)

connected to an HPLC system (Shimadzu Scientific) in an aqueous solution of ammonium phosphate (20–50 mM) at a pH of 6.0 using a gradient of sodium chloride up to 0.5 M. Products were detected by fluorescence of 4-methylumbelliferone using an excitation wavelength 325 nm and an emission wavelength 380 nm.

### SDS-PAGE
Proteins were resolved on homemade 3–15% polyacrylamide gradient gels in SDS-glycine buffer for either 6 h at 200 V or 16 h at 60 V at room temperature. The proteins were transferred to the PVDF membrane at 800 mA for 5 h at 4 °C.

### Western blotting
Membranes were blocked in either 2% skim milk in 50 mM Tris pH 7.6 and 75 mM NaCl with 0.1% TWEEN 20 (low-salt TBS-T) or fish gelatin dissolved in low-salt TBS-T and incubated with antibodies against matriglycan (IIH6), DGN (1D9) or dystroglycan (AF6868) for 16 h at 4 °C. For laminin overlays, membranes were blocked in 5% skim milk dissolved in 10 mM ethanolamine pH 7.6, 140 mM NaCl, 1 mM $MgCl_2$, and 1 mM $CaCl_2$. The membranes were overlaid with mouse laminin (ThermoFisher Scientific) in 3% BSA dissolved in the same buffer and incubated for 16 h. The membranes were washed and incubated with rabbit anti-laminin antibody (Sigma-Aldrich) for 16 h at 4 °C. Appropriate infra-red fluorescence-conjugated secondary antibodies were used in corresponding blotting buffers. The membranes were visualized on a Li-COR Odyssey CLx.

### Quantification and statistical analysis
Statistical analyses for method-specific experiments: MALS, SAXS, and cryo-EM reconstruction data are presented. SEC-MALS-SAXS experiments have been performed multiple times ($n > 3$) over five years with corresponding results. Matriglycan polymerization probed by Western blots was performed in triplicate unless otherwise noted.

### Animals
Animal care, ethical usage, and procedures were performed in strict accordance with protocols approved by the National Institutes of Health and the Institutional Animal Care Use and Committee (IACUC). Mice were socially housed (unless single housing was required) under specific-pathogen-free conditions in an Association for Assessment and Accreditation of Laboratory Animal Care (AAALAC)-accredited animal facility. Mouse housing conditions were as specified in the Guide for the Care and Use of Laboratory Animals (National Research Council).

### T190M mutant mice
Prodystroglycan T190M knock-in mouse model, tissue homogenization and WGA affinity preparation was previously described by Hara et al.[28].

### AlphaFold 3 models of LARGE1-prodystroglycan
AlphaFold 3 server was used to generate five models using two copies of LARGE1 sequence (residues 31-756), with tetra-antennary N-glycan modifications (92N, 242N, 724N and 725N with glycan chain NAG(FUC)(NAG(MAN(MAN(NAG)(NAG))(NAG)))) and a single chain of dystroglycan (28-320) along with four $Mn^{2+}$ ions and one $Ca^{2+}$ ion. Models with and without N-glycans were also generated (Supplementary Fig. 18). The random seeds were 736608798, 1561488507 and 1860396058 for the model without N-glycans. A model of matriglycan was manually placed in the cleft of DGN to judge degree of polymerization.

### Reporting summary
Further information on research design is available in the Nature Portfolio Reporting Summary linked to this article.

## Data availability
Unless otherwise stated, all data supporting the results of this study can be found in the article, supplementary, and source data files. Source data are provided with this paper. LARGE1 coordinates and maps have been deposited with the accession codes 7UI6 and EMD-26540 (C1 symmetry: [https://www.ebi.ac.uk/emdb/EMD-26540]) and 7UI7 and EMD-26541 (C2 symmetry: [https://www.ebi.ac.uk/emdb/EMD-26541]). The map and model for LARGE1dTM with DGN are also deposited in the PDB under the accession code 9E1T and EMD-47420. Glycoproteomic mass spectrometric data for LARGE1dTM have been deposited in PRIDE database under accession code PXD060053. SAXS data have been deposited in the SASBDB repository accession code SASDNF8, accession code SASDNG8, accession code SASDNH8, accession code SASDNJ8. Source data are provided with this paper.

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

## Acknowledgements

This manuscript is the result of funding in whole or in part by the National Institutes of Health (NIH). It is subject to the NIH Public Access Policy. Through acceptance of this federal funding, NIH has been given a right to make this manuscript publicly available in PubMed Central upon the Official Date of Publication, as defined by the NIH. This research used resources from the Advanced Photon Source, a U.S. Department of Energy (DOE) Office of Science User Facility operated for the DOE Office of Science by Argonne National Laboratory under Contract No. DE-AC02-06CH11357. J.H. was supported by grants P41 GM103622 and P30 GM138395 from the National Institute of General Medical Sciences (NIGMS) of the National Institutes of Health (NIH). J.H. provided the use of the Pilatus 3 1M detector which was funded by grant 1S10OD018090 from NIGMS. The content is solely the responsibility of the authors and does not necessarily reflect the official views of the NIGMS or the NIH. Extraordinary facility operations were supported in part by the DOE Office of Science through the National Virtual Biotechnology Laboratory, a consortium of DOE national laboratories focused on the response to COVID-19, with funding provided by the Coronavirus Aid, Relief, and Economic Security (CARES) Act. A portion of this research was supported by O.D. through NIH grant U24GM129547 and performed at the Pacific Northwest Center for Cryo-EM (PNCC) at Oregon Health & Science University (OHSU) and accessed through Environmental Molecular Sciences Laboratory (EMSL; grid.436923.9), a DOE Office of Science User Facility sponsored by the Office of Biological and Environmental Research. Research reported in this publication was supported by the National Institute of Neurological Disorders and Stroke of the National Institutes of Health under Award Number P50 NS053672 (K.P.C.). The content is solely the responsibility of the authors and does not necessarily represent the official views of the NIH. We sincerely thank Dr. Jennifer Barr of the Scientific Editing and Research Communication Core at the University of Iowa Carver College of Medicine for her critical review of the manuscript. We also appreciate the administrative support provided by Amber Mower and Jaeda Harmon. We thank Drs. Sandipan Chowdhury, Erhard Hohenester, Andrew Ward, Lance Wells, and Liping Yu for their constructive discussions and input.

## Author contributions

K.P.C. and S.J. conceptualized the study. S.J., N.J.S., N.S., Z.X., T.Y., O.D., J.H., M.E.A., D.V. curated the data. S.J., N.J.S., N.S., Z.X., T.Y., M.E.A., D.V. performed formal analysis. K.P.C. funded the research. S.J., K.P.C., N.J.S., N.S., Z.X., R.Y., Z.Y., T.Y., O.D., J.H., M.E.A., D.V. conceived the investigation. S.J., K.P.C., N.J.S., N.S., Z.X., R.Y., Z.Y., T.Y., O.D., J.H., M.E.A., D.V. conceived the methodology. S.J. was the project administrator. N.J.S., N.S., Z.X., T.Y., O.D., J.H. provided resources. K.P.C. supervised the study. S.J., N.J.S., N.S., Z.X., T.Y., O.D., J.H., M.E.A., D.V. validated the data. S.J., N.J.S., Z.X., N.S., T.Y., M.E.A., D.V. visualized the project. S.J., K.P.C. wrote the original draft of the manuscript. S.J., K.P.C., N.J.S., Z.X., T.Y., J.H. reviewed and edited the manuscript.

## Competing interests

The authors declare no competing interests.
