## [Transparent Peer Review file · Nature Communications]

LARGE1 processively polymerizes length-controlled matriglycan on prodystroglycan

Corresponding Author: Professor Kevin Campbell

Version 0:

Reviewer comments:

Reviewer #1

(Remarks to the Author)

In this paper by Joseph S. et al., the authors present a structural and biochemical analysis of matriglycan synthesis by LARGE1 and discover a role for the dystroglycan N-terminal domain in controlling the length of the polymer. Overall, this is a well-conducted, important, and informative study that provides new mechanistic data for matriglycan synthesis and will significantly contribute to the field. Some additional work on the clarity and data presentation will likely improve the manuscript.

Major comments:

A major argument that the authors make is that in-vivo, LARGE1 synthesizes matriglycan at a discrete size due to a “ruler” function of DGN. The tissue analysis in Figure 5e shows a fairly large smear, indicating a heterogenous and not discrete size of matriglycan. How is this observation reconciled with the claims about a discrete size?

The authors should try to improve the narrative of this paper, as the motivation, especially in light of previously published structural data, is not always so clear.

Table S2 refers to 7UI6 and 7UI7, the data for 7E1T is missing. Also, for all structures, the complete PDB validation reports need to be provided.

The authors state that in the presence of the DGN they observed UDP in the active site glucuronic acid transferase. A supplementary figure showing the density of the UDP needs to be presented.

The authors present an AlphaFold-generated model for the LARGE1/DGN complex. First, there isn't any description of the prediction in the material and methods. Was a single model generated or multiple ones? Based on accumulated experience, AlphaFold can sometimes produce very different models for the same target proteins. The authors should perform the prediction several times and report if the same solution is generated or not. Also, the authors should provide a supplementary figure to show the confidence level that AlphaFold generates. Furthermore, how matriglycan was modeled into the cleft on DGN? The authors need to provide additional details and some metrics for the goodness of fit.

Minor points:

It looks like some of the references throughout the manuscript may have gotten scrambled; the authors should double-check that.

In Figure 1a, there is a large empty white space that can be eliminated by adjusting the Y-axis scale. Also, the authors state that the peaks follow a Poisson distribution. A fit to such distribution should be shown either in Figure 1a or in a supplementary figure to back up this claim.

In line 137, multi-angle light scattering and not as written.

The supplementary figures need to be reorganized as they are not called in the order they appear (specifically the SAXS-related figures).

In Table S2, the magnification is suspiciously low. The authors should double-check this value.

Please add a legend to Figure 3 to specify the sugar identity of each of the graphical symbols presented.

Supplementary figures 7 & 8 should be revised to provide a better description of the data processing pipeline, including the exact number of particles in each step, FSC curves, and orientational distribution when appropriate.

In Figure 5e, the MW marker should be annotated.

Reviewer #2

(Remarks to the Author)

This study provides significant structural insights into the dimerization and catalytic mechanisms of LARGE1, with implications for understanding dystroglycanopathies. While the findings are significant, several areas require optimization and elaboration to improve the impact and scientific rigor of the manuscript.

1. Integrate SAXS data using CRY SOL, FoXS, and Multi-FoXS.
2. Provide a comprehensive comparison of LARGE1 and LARGE2, addressing the rationale for LARGE2's limited discussion.
3. Emphasize novel contributions over prior work, esp. ref. 40, particularly in the absence of the LARGE1–prodystroglycan complex.
4. Include the missing ITC figure and address minor inconsistencies in figure numbering and supplementary materials.

1. Fitting Cryo-EM – obtained structures into SAXS Profiles

The inclusion of cryo-EM data is a strength of the study; however, its integration with SAXS data could bolster the structural validation:

- CRY SOL: Using CRY SOL to fit the LARGE1 cryo-EM-obtained dimer structure into SAXS data would quantitatively evaluate the consistency of the structural model with the experimental scattering curves. This analysis should include metrics such as chi-square (χ^2), p-values, and weighted residuals.
- FoXS and Multi-FoXS: To further validate the SAXS fitting, FoXS and Multi-FoXS could be employed. These tools provide a comparative analysis of chi-square values and assess the robustness of the structural model against potential conformational heterogeneity in solution.

2. Sequence and Functional Comparison of LARGE1 and LARGE2

A detailed comparison between LARGE1 and LARGE2, two homologous glycosyltransferases, is notably absent but essential for contextualizing the study:

- Sequence Analysis: LARGE1 (UniProt ID:O95461) and LARGE2 (UniProt ID:Q8N3Y3) share similar catalytic roles but differ in sequence and likely function. A comparative analysis of key domains, conserved motifs, and divergence in active site residues would strengthen the study.
- Functional Context: LARGE2 is mentioned briefly in the SAXS experiments but is otherwise omitted from the discussion. Addressing why LARGE2 was included in SAXS but not in other structural analyses is critical. Does it differ in dimerization behavior, stability, or catalytic efficiency? Clarifying its relevance will provide a more complete picture.
- Rationale for LARGE2 Omission: If LARGE2 was excluded due to technical constraints (e.g., expression or purification issues), this should be explicitly stated. Otherwise, a discussion of its biological significance and structural implications is necessary.

3. Missing LARGE1–prodystroglycan Complex and Novelty Assessment

The absence of the LARGE1–prodystroglycan complex, previously deposited with PDB ID 7E1T and EMD-47420, limits the study's scope:

- Comparison with 7ZVJ: The authors should clearly delineate how their findings build upon prior work by Katz and Diskin. For example, the discussion of dimerization dynamics should explain the relevance of anti-parallel dimers as a potential artefact (or not) of the omission of the CC domain (ref. 40). While the dimerization dynamics is explored in greater detail, emphasizing new insights—such as substrate retention mechanisms, catalytic proximity effects, or disease mutation impacts—would enhance the novelty.
- Highlighting Contributions: Without the prodystroglycan complex, the work risks being perceived as incremental. Explicitly highlighting unique contributions, such as the SAXS data integration, the impact of dimerization on activity, or the interpretation of disease-relevant mutations, would mitigate this concern.
- Target Journal: Given the absence of the prodystroglycan complex, the study may not meet the high novelty bar of Nature Communications. It would be better positioned in a specialized journal focused on structural biology or enzymology.

4. ITC Data Inclusion

The manuscript includes data and methods for isothermal titration calorimetry (ITC), but the corresponding figure is missing. Including this figure would substantiate claims about the binding interactions and thermodynamics of substrate recognition. The absence of this figure detracts from the completeness of the study and should be addressed promptly.

5. Minor Issues and Recommendations

- Figure Numbering and Supplementary Data: Correct figure mislabeling (e.g., Figure 1c, 10-14 should be Figure 1c, 9-13 in the supplementary section) to ensure clarity and consistency.
- Writing Style: Several sections of the manuscript are verbose and could be streamlined.
- Structural Comparisons: Adding comparisons to other glycosyltransferases, such as LARGE2 or related enzymes, would situate the findings within the broader context of glycosylation research.

Reviewer #3

(Remarks to the Author)

I co-reviewed this manuscript with one of the reviewers who provided the listed reports. This is part of the Nature Communications initiative to facilitate training in peer review and to provide appropriate recognition for Early Career

Researchers who co-review manuscripts.

Reviewer #4

(Remarks to the Author)

Matriglycan, a post-translational moiety on dystroglycan, plays important biological functions through binding with various ligands, for example, physical maintenance of skeletal muscle fibers, brain development, and Lassa fever virus infection. Mutations in proteins involved in matriglycan modification cause muscular dystrophy and lissencephaly. Although dystroglycan gene DAG1 and genes involved in matriglycan modification are ubiquitously expressed, matriglycan length (i.e., ligand binding activity) varies depending on the tissue and the cell type. However, it is unknown how matriglycan length is controlled. Elucidation of this question will lead to a deeper understanding of the mechanisms of glycosylation and muscular dystrophy. The authors propose a new mechanism of matriglycan modification through experiments in structural biology, biophysics, biochemistry, etc. However, there is insufficient data to draw the conclusion and proposal that the authors claim. Otherwise, the authors should weaken the claim or add more data. Some concerns are described below.

Throughout this article, the authors refer to the synthesis style of matriglycan as “processively” or “distributively”, but the meaning and definition of these terms should be clearly explained. I (general readers as well) may not have fully understood what the authors claim.

For in vitro matriglycan synthesis assay (Fig 4d, right gel), how did the authors make the DXD1:DXD3 heterodimer? Since the authors mentioned that there is no exchange between different homodimers in the solution, I understand that they did not mix separately purified Dx1 and DXD3. If they made it by co-transfection, the preparation would contain not only heterodimers but also homodimers (theoretically, DXD1 homo, heterodimer, and DXD3 homo are 1:2:1). In that case, wouldn't matriglycan synthesis proceed by the two types of homodimers contained in the preparation? This question is also for pseudomeric LARGE1dTM (Dead-Live heterodimer).

In relation to the above comments, they said that there is no exchange because each homodimer of DXD1 and DXD3 is SDS-resistant, but this does not prove that exchange does not occur when DXD1 homodimer and DXD3 homodimer are mixed in solution. For example, why not mix homodimers tagged with different tags, such as DXD1-myc and DXD3-FLAG, and perform immunoprecipitation with one of the tags?

In the right gel of Fig. 4d, why does wildtype LARGE1dTM not produce a smear band of high molecular weight after a long-time reaction? As I mentioned before, this may depend on the definition of “processively” and “distributively”. If “processively” means extending matriglycan, then shouldn't wildtype LARGE1dTM produce a smear band that spreads to high molecular weight?

As shown in Fig. 5 and Supplementary Fig. 6, the T192M mutant is modified with matriglycan in HEK cells and mouse tissues, and its binding to LARGE1 and affinity of UDP-GlcA to the ES complex are comparable to those of the wild type. However, it is very puzzling why in vitro polymerization does not occur at all (Supplementary Fig. 6b and c). The authors speculate that there is a different factor in vivo, but if so, the data is insufficient to claim that DGN regulates the length of matriglycan. In addition, this contradicts the results of a previous report from the same group (New Engl J Med 2011) that the T192M mutation disrupts LARGE-dystroglycan interaction.

The proposal that DGN functions as a molecular ruler is very interesting, but the only data leading to this proposal is dry data from Caver analysis and alphaFold3 prediction. Even if DGN functions as a molecular ruler, it cannot explain the difference in matriglycan length between cell types and tissues in wild-type mice. It also cannot explain the phenomenon that the WT band is extended to a smear high molecular weight by overexpression of LARGE.

There are some parts where the description of the experimental method is insufficient and the results cannot be followed accurately. Specifically; Is the prodystroglycan a sample of 100mM DEAE elution; How was the LARGE1dTM-DAG1 reconstituted used for cryoEM and UDP titration assay?

In the schematic diagram of the glycan structure in Figure 1C, the glycan of prodystroglycan stops at xylose-glucuronic acid, just before matriglycan modification. This is probably true because matriglycan polymerization does not occur after b-GUS treatment as in the experiment in Figure 3, but there is no direct evidence that the schematic diagram is correct because there are no glycan analysis experiments. It would be better to include data showing that matriglycan modification does not occur in prodystroglycan expressed in B4GAT1-KO cells though it is also indirect evidence.

The following points need to be checked for accuracy and readability.

Lines 121 and 140, DAG1 28-340. Is 28-340 correct? In Fig2e it says 28-749.

Lines 401 and 403, DAG1 28-340, 28-398. Are these correct?

The reference numbers in the text do not match the correct reference in the list.

Version 1:

Reviewer comments:

Reviewer #1

(Remarks to the Author)

The authors have addressed all my comments.

I highly recommend accepting this revised manuscript for publication.

Reviewer #2

(Remarks to the Author)

Reviewer 2 – Second Round Report

I have read the corrected manuscript, the complete supplementary material, and the point-by-point rebuttal. The authors have invested substantial effort, and many of my earlier technical concerns are now addressed. The paper is clearer, and the new supplementary figures (especially the SAXS fits and AlphaFold confidence maps) are helpful. A few scientific and presentation issues still need attention before the work is ready for publication.

1 Overall assessment

The study tackles a genuinely important question—how LARGE1 sets matriglycan length on dystroglycan—and combines cryo EM, SAXS, mutagenesis and enzymology rigorously. The conclusion that LARGE1 is processive on native prodystroglycan yet distributive on small substrates is well supported. The further suggestion that the DGN domain limits polymer length is plausible and interesting; however, the mechanistic link between DGN, enzyme dimer geometry, and chain length remains speculative.

2 Major comments

Comment Suggested action

2.1 “Molecular ruler” language still too categorical. Figures 5f and Supplementary 18 rely on AlphaFold docking, and there is no direct density for DGN in the catalytic cleft. Re phrase throughout to make it explicit that the ruler model is hypothetical. E.g. change “DGN controls matriglycan length by acting as a ruler” → “our data are consistent with a ruler like role for DGN, although the structural details (Figs S18c are very small) remain to be elucidated.” It would also be helpful to label the respective XylT and GlcAT domains and catalytic site on the AF model of the complex (Fig S18d).

2.2 T192M: loss of O glycosylation vs. steric clash. Thr 192 could itself be O glycosylated; replacing it with Met might shorten matriglycan indirectly by removing that glycan. The rebuttal mentions this, but the manuscript does not. Add a short paragraph in the Discussion acknowledging this alternative explanation and, if possible, cite any (MS) data or prediction (eg. NetOGlyc) indicating whether or not Thr192 may carry an O glycan in WT dystroglycan.

2.3 Connection between dimer asymmetry and length control is still vague. One xylosyl transferase domain is poorly resolved, but the functional implications are unclear. Add 1–2 sentences clarifying whether the authors think asymmetry itself gates processivity or is simply a cryo EM artefact, to aid non structural readers.

2.4 LARGE1 vs. LARGE2 context. A sequence alignment is now provided (Supp. Fig. 4), but functional differences between the paralogues are not discussed. Insert 2–3 sentences in the Introduction or Discussion summarising known LARGE2 activity and why it was not pursued here.

2.5 Proposed N glycan structure lacks the conserved trimannose core. A dimannose core would be highly atypical for a complex glycan. Provide experimental evidence (e.g., MS/MS, NMR, targeted exoglycosidase digestion) to support the assignment; clarify whether the glycan occurs naturally or is synthetic; outline a plausible biosynthetic route; and confirm that the omission of the two mannose residues is not a typographical error.

2.6 UDP GlcA only MST assay: the primer ends in GlcA, so UDP GlcA could bind non productively; absence of a UDP Xyl control leaves the 30 fold affinity increase open to alternative interpretation. Supply MST data (or a rationale) for: (i) pre incubation with UDP Xyl to generate a xylose terminated acceptor, and/or (ii) simultaneous UDP Xyl + UDP GlcA conditions; explain how these controls support the current Kd interpretation.

The ITC data (about 1,2 uM Kd) on Large1 dTM-DAG1 titration with GlcA may indicate a displacement event (this would be in line with a processive transfer action) that might be fitted using a competitive binding model using more advanced software (eg. AFFINIMETER)?

2.7 Supplementary Fig. 4 (line 122 p. 6)

is Fig S3 in the supplementary materials and only shows part of the GlcA monosaccharide part. Correct and remediate by showing the whole UDP-GlcA donor sugar, similar to the zoom-in in Fig 2d.

3 Minor comments

1. Figure 1a – The Poisson fit is now in Supp. Fig. 1; please reference it explicitly in the legend. The DAG1 construct used to make the complex with Large1 dTM needs to be called like that in the Figure (1b): is DAG1 the second structure (Golgi, with DGN) or third structure (cytosolic, upon cleavage by furin)?

2. Figure 3 legend – good addition of sugar symbols, but please define “Xyl GlcA” directly in the caption for first-time readers.

3. Supplementary Fig. 11 – include χ^2 values in the panel or caption to make goodness of fit immediately visible.

4. Supplementary Fig. 16 – Explain what is C1 and C2 symmetry: simply introducing a 2-fold or more (like C-centering?). Add PDB codes of obtained structures in the legend.

5. Supplementary Fig. 17 – Large1 is bound to more than UDP along, or can this not be inferred? Please add the PDB codes for the obtained structures.

6. Typos –

1. line 302, “biding” should be “binding.”

2. Line 239 active sites on one protomer is -> are 239 mutated

3. Line 309 “arena viral infection” should be “arenaviral infection”

4. Caption Fig 5f – “Threonine 192 is show in red” → “is shown”.
5. Line 453 - “SEC MALS SAX was performed” should be “SEC MALS SAXS”.

Reviewer #3

(Remarks to the Author)

Reviewer #4

(Remarks to the Author)

My concerns have been appropriately addressed in this revision. It would be very helpful for readers to include the following points; 1) methods to obtain heterodimers of DXD1-DXD3 as the authors explained in response #2, and 2) a possible explanation why wt LARGE1dTM did not produce a smear band of high mw after longer incubation as the authors explained in response #4, i.e., “A processive enzyme may not extend polymers ad infinitum. Many processive polymerases rely on an orthogonal polymeric template to govern the length of the newly synthesized polymer, but other length-controlling mechanisms may be used in the case of template-free syntheses”.

Version 2:

Reviewer comments:

Reviewer #2

(Remarks to the Author)

All comments were taken into account.

Point-by-Point Response to Reviewer Comments

We have thoroughly amended our manuscript, including figures, according to the comments from the reviewers.

(Please see our responses in blue italics)

Reviewer 1

Remarks to the Author:

In this paper by Joseph S. et al., the authors present a structural and biochemical analysis of matriglycan synthesis by LARGE1 and discover a role for the dystroglycan N-terminal domain in controlling the length of the polymer. Overall, this is a well-conducted, important, and informative study that provides new mechanistic data for matriglycan synthesis and will significantly contribute to the field. Some additional work on the clarity and data presentation will likely improve the manuscript.

Major Comments:

1. A major argument that the authors make is that in-vivo, LARGE1 synthesizes matriglycan at a discrete size due to a “ruler” function of DGN. The tissue analysis in Figure 5e shows a fairly large smear, indicating a heterogenous and not discrete size of matriglycan. How is this observation reconciled with the claims about a discrete size?

Thank you for pointing out the discrepancy between our wording and the image of the band on WB. We agree that tissue analysis shows a large smear. However, the size of the smear is discrete for each tissue (for example, 100 kDa to 120 kDa for brain, 150 kDa to 170 kDa for skeletal muscle). The mucin-like domain carries other O-glycans that contribute to the smear, which are heterogeneously appended to dystroglycan and differ between tissues.

To reconcile our claims, we have 1) appended Figure 1b (schematic of prodystroglycan) with asterisks to indicate extensive O-glycosylation and modified the figure legend. 2) added a phrase in red and a reference in the introduction to indicate that dystroglycan is heavily O-glycosylated. 3) Figure 5e shows that in all examined organ types, the bands from T192M mouse extracts exhibit faster migration, indicating a lower apparent molecular weight compared to the corresponding wild-type samples. You may be expecting a tight band by the word “discrete”; we have therefore also changed the wording to “discrete distribution” to indicate a migrating front and well-defined termination point. 4) The thickness (from migration front to end) holds true for the distribution of recombinant prodystroglycan without matriglycan, observed in Figure 1d (top panel, 100 mM DEAE eluate stained with Coomassie brilliant blue, and third panel, anti-DGN (1D9) Western blot) and in Figure 1e, distribution of core dystroglycan without matriglycan (right-most panel), suggesting that it may not be matriglycan that is responsible for the band thickness.

2. The authors should try to improve the narrative of this paper, as the motivation, especially considering previously published structural data, is not always so clear.

We have provided the rationale in red in the revised manuscript (lines 69-70)

Considering the previously published LARGE1 structure, we describe in our abstract that 1) we could not decipher the mechanism of matriglycan polymerization from the cryo-EM reconstruction of LARGE1 alone and thus needed to conduct biochemical/enzymatic experiments. This required a proteinaceous native-like substrate that would control matriglycan length rather than the synthetic 4-methylumbelliferone-conjugated substrate. 2) The manuscript title describes a processive length-controlled mechanism of matriglycan synthesis that cannot be gleaned from the cryo-EM reconstruction of LARGE1 and is absent in Katz & Diskin 2022.

3. Table S2 refers to 7UI6 and 7UI7, the data for 9E1T is missing.

The data for 9E1T is appended to the table.

4. Also, for all structures, the complete PDB validation reports need to be provided.

*The complete PDB validation reports are provided as a separate PDF and *.cif.gz files. These are also available directly from the PDB since the entries have been released.*

5. The authors state that in the presence of the DGN they observed UDP in the active site glucuronic acid transferase. A supplementary figure showing the density of the UDP needs to be presented.

The density of the UDP is presented in Supplementary Figure 3.

6. The authors present an AlphaFold-generated model for the LARGE1/DGN complex. First, there isn't any description of the prediction in the material and methods.

Thank you for pointing this out, and we apologize for the omission. We have added the section "AlphaFold 3 models of LARGE1-prodystroglycan" to the Methods.

7. Was a single model generated or multiple ones? Based on accumulated experience, AlphaFold can sometimes produce very different models for the same target proteins.

In our studies, AlphaFold 3 generated multiple models, which are presented in Supplementary Figure 18.

8. The authors should perform the prediction several times and report if the same solution is generated or not.

This has been completed. We have performed the AlphaFold 3 prediction three times with random seeds. One of those three times was without any N-glycan showing that the binding

interface of DGN remains constant. We added Supplementary Figure 18 and appended the Methods section with the random seeds (lines 611-12).

9. Also, the authors should provide a supplementary figure to show the confidence level that AlphaFold generates.

We added Supplementary Figure 18 to the Supplementary Materials. This figure shows the different AlphaFold 3 models of LARGE1 with DAG1, with the confidence reported in panel g.

10. Furthermore, how matriglycan was modeled into the cleft on DGN? The authors need to provide additional details and some metrics for the goodness of fit.

The model is hypothetically based on the location of the T192M mutation. We manually placed a chain of matriglycan in the cleft to estimate its degree of polymerization in Figure 5f. We added this information to the Methods section to clarify this point (lines 603-5).

Minor Points:

1. It looks like some of the references throughout the manuscript may have gotten scrambled; the authors should double-check that.

We apologize for any confusion this may have caused. Reference numbering throughout the manuscript has been corrected.

2. In Figure 1a, there is a large empty white space that can be eliminated by adjusting the Y-axis scale.

The spacing in Figure 1a has been adjusted.

3. Also, the authors state that the peaks follow a Poisson distribution. A fit to such distribution should be shown either in Figure 1a or in a supplementary figure to back up this claim.

We have added this information to Supplementary Figure 1.

4. In line 137, multi-angle light scattering and not as written.

We changed the wording to “light” scattering in Supplementary Figure 2b.

5. The supplementary figures need to be reorganized as they are not called in the order they appear (specifically the SAXS-related figures).

The supplementary figures have been correctly sequenced.

6. In Table S2, the magnification is suspiciously low. The authors should double-check this value.

The data were collected on a microscope without an energy filter. Thus, the values are lower than those observed using newer microscopes that have energy filters.

7. Please add a legend to Figure 3 to specify the sugar identity of each of the graphical symbols presented.

A legend defining the graphical symbols has been added to Figure 3.

8. Supplementary figures 7 & 8 should be revised to provide a better description of the data processing pipeline, including the exact number of particles in each step, FSC curves, and orientational distribution when appropriate.

The legends for Supplementary Figures 7 (changed to S16) and 8 (changed to S17) have been updated to include more information about the data processing pipeline.

9. In Figure 5e, the MW marker should be annotated.

The MW marker has been added to Figure 5e.

Reviewer 2

Remarks to the Author:

This study provides significant structural insights into the dimerization and catalytic mechanisms of LARGE1, with implications for understanding dystroglycanopathies. While the findings are significant, several areas require optimization and elaboration to improve the impact and scientific rigor of the manuscript.

Major Comments:

1. Integrate SAXS data using CRY SOL, FoXS, and Multi-FoXS.

Supplementary Figure 11 has been added to address these points.

2. Provide a comprehensive comparison of LARGE1 and LARGE2, addressing the rationale for LARGE2's limited discussion.

We have provided a sequence comparison between human LARGE1 and LARGE2 concerning the absence of a coiled-coil domain in the latter in Supplementary Figure 4, but further discussion of LARGE2 is beyond the scope of this paper.

3. Emphasize novel contributions over prior work, esp. ref. 40, particularly in the absence of the LARGE1–prodystroglycan complex.

The majority of our results (Figures 1, 3, 4, and 5) address whether LARGE1 is processive or distributive and controls the length of matriglycan, not its structure. The table emphasizes the novel contributions of our work over prior work.

Overlap between Diskin et al and the current publication	Katz et al	Joseph et al
Produce recombinant proteinaceous substrate for LARGE1 that recapitulates matriglycan synthesis in vivo	no	yes
Uses 4-methylumbelliferone for matriglycan synthesis to show that matriglycan polymers follow a Poisson distribution	no	yes
Reconstructs cryo-EM structures of LARGE1 constructs	yes	yes
Solves X-ray crystal structures of LARGE1 constructs	yes	no
Analyze LARGE1 dimer interface	yes	minimal
Analyze expression of LARGE1 pathological mutations	yes	no
Maps patient mutations onto structure of LARGE1	yes	no
Hypothesize that matriglycan is polymerized by alternating active sites on the two protomers	yes	yes
Models substrate into apo-LARGE1 active sites from existing structures	yes	no
Has density for UDP-Glucuronate in LARGE1-prodystroglycan enzyme-substrate reconstruction	no	yes
Shows LARGE1 dimerizes in solution without coiled-coil domain	yes	yes
Shows that LARGE1 dimer has dynamic asymmetry using C1 reconstruction	no	yes
Shows LARGE1 coiled-coil stem domain using SAXS Has SAXS analysis of LARGE1 and LARGE2 with and without N-glycans	no	yes
Has N-glycan analysis of LARGE1 by mass spectrometry	no	yes
PRE-REQUISITES and MECHANISM of matriglycan synthesis and LENGTH control		
Shows that LARGE1 efficiently polymerizes matriglycan on prodystroglycan only when modified with phosphorylated core M3	no	yes
Shows that DGN is necessary of LARGE1 matriglycan synthesis activity on (pro)dystroglycan	no	yes
Shows that a xylose-glucuronate primer is necessary for LARGE1 matriglycan synthesis on prodystroglycan	no	yes
Shows that LARGE1 processively polymerizes matriglycan on prodystroglycan but does so distributively on the synthetic substrate 4-methylumbelliferone-glucuronate-xylose	no	yes
Shows that pseudomonomeric or half-dead LARGE1 enzymes cannot polymerize matriglycan	no	yes
Shows that UDP and UDP-glucuronate only bind to the LARGE1-prodystroglycan enzyme-substrate complex	no	yes
Shows that the patient mutations in prodystroglycan T192M decreases the length of matriglycan polymerized	no	yes
Hypothesizes that length control of matriglycan by LARGE1 relies on a cleft in DGN	no	yes
Shows that prodystroglycan T192M binds LARGE1 and can form an active enzyme-substrate complex capable of bind UDP-glucuronate but cannot transfer sugars	no	yes

4. Include the missing ITC figure and address minor inconsistencies in figure numbering and supplementary materials.

The ITC result has been omitted from the manuscript because of the extremely shallow binding curve without a clear point of inflexion that identifies stoichiometry. Instead, we included an explanation in point #11 of this point-by-point response document.

5. Fitting Cryo-EM – obtained structures into SAXS Profiles. The inclusion of cryo-EM data is a strength of the study; however, its integration with SAXS data could bolster the structural validation:
 - CRY SOL: Using CRY SOL to fit the LARGE1 cryo-EM-obtained dimer structure into SAXS data would quantitatively evaluate the consistency of the structural model with the experimental scattering curves. This analysis should include metrics such as chi-square (χ^2), p-values, and weighted residuals.
 - FoXS and Multi-FoXS: To further validate the SAXS fitting, FoXS and Multi-FoXS could be employed. These tools provide a comparative analysis of chi-square values and assess the robustness of the structural model against potential conformational heterogeneity in solution.

We fitted the cryo-EM model to the SAXS profiles using CRY SOL and FoXS. We included goodness of fit and residuals. Lines 139-40 in the manuscript and Supplementary Figure 11 have been added to address these points. Deciphering the mechanism of matriglycan synthesis and length control is the focus of this study.

6. Sequence and Functional Comparison of LARGE1 and LARGE2

A sequence comparison is provided in the supplementary section.

7. A detailed comparison between LARGE1 and LARGE2, two homologous glycosyltransferases, is notably absent but essential for contextualizing the study:

A detailed study and comparison of LARGE2 with LARGE1 is beyond the scope of this study and may form the basis of another manuscript.

- Sequence Analysis: LARGE1 (UniProt ID:O95461) and LARGE2 (UniProt ID:Q8N3Y3) share similar catalytic roles but differ in sequence and likely function. A comparative analysis of key domains, conserved motifs, and divergence in active site residues would strengthen the study.
- Functional Context: LARGE2 is mentioned briefly in the SAXS experiments but is otherwise omitted from the discussion. Addressing why LARGE2 was included in SAXS but not in other structural analyses is critical. Does it differ in dimerization behavior, stability, or catalytic efficiency? Clarifying its relevance will provide a more complete picture.

We added that LARGE2dTM was used in the SEC-MALS-SAXS studies to show that dimerization occurs in the absence of a coiled-coil domain, which is only present in LARGE1(dTM) in lines 138-42.

8. Rationale for LARGE2 Omission: If LARGE2 was excluded due to technical constraints (e.g., expression or purification issues), this should be explicitly stated. Otherwise, a discussion of its biological significance and structural implications is necessary.

LARGE2dTM doesn't express nearly as well as LARGE1, but that's not the reason for its exclusion. LARGE2dTM was primarily used to determine whether the coiled-coil of LARGE1 was required for dimerization (Figure 2, Supplementary Figure 2&3). This paper is limited to the mechanism and length control of matriglycan synthesis by LARGE1 on prodystroglycan. Analyzing LARGE2 would form a separate manuscript.

9. Missing LARGE1–prodystroglycan Complex and Novelty Assessment.

The absence of the LARGE1–prodystroglycan complex, previously deposited with PDB ID 7E1T and EMD-47420, limits the study's scope:

- Comparison with 7ZVJ: The authors should clearly delineate how their findings build upon prior work by Katz and Diskin. For example, the discussion of dimerization dynamics should explain the relevance of anti-parallel dimers as a potential artefact (or not) of the omission of the CC domain (ref. 40). While the dimerization dynamics is explored in greater detail, emphasizing new insights—such as substrate retention mechanisms, catalytic proximity effects, or disease mutation impacts—would enhance the novelty.

Topics like dimerization and disease mutations, as discussed by Katz and Diskin, are not revisited here to avoid redundancy. The mechanism of LARGE1 matriglycan polymerization does not build on Katz and Diskin's work because they never performed enzymatic assays. We specifically do not address dimerization dynamics because Katz and Diskin have addressed it. We do not use constructs of LARGE1 without the coiled-coil domain because they are not biologically relevant for understanding the mechanism of matriglycan polymerization. Thus, we do not make a comment on the anti-parallel confirmation of LARGE1 without a coiled coil domain. Katz and Diskin used constructs of LARGE1 without a coiled-coil domain for ease of crystallization, which is not the purpose of this work. Our paper's focus is on the processivity of matriglycan synthesis by LARGE1 on prodystroglycan and its length control, as reflected in the title of the manuscript and shown by figures 1, 3-5. Please see the provided comparative table to highlight the distinctions between our publications and those of Katz and Diskin. We do not have data on substrate retention or catalytic proximity effects.

- Highlighting Contributions: Without the prodystroglycan complex, the work risks being perceived as incremental

As indicated in the paper's title, our work focuses on the processivity of matriglycan synthesis and its length control, neither of which has been previously explored. Please reference the provided comparative table to highlight the distinctions between our publications and those of Katz and Diskin.

10. Explicitly highlighting unique contributions, such as the SAXS data integration, the impact of dimerization on activity, or the interpretation of disease-relevant mutations, would mitigate this concern.

The unique contributions of this work explore processive matriglycan synthesis and length control, which cannot be gleaned from the structure alone – please see the provided comparative table to highlight the distinctions between our publications and those of Katz and Diskin. The impact of dimerization on activity cannot be assessed because LARGE1 appears to be an obligate dimer, given that pseudomeric LARGE1 has no matriglycan synthesis activity. We do not address disease-relevant mutations because Katz and Diskin already addressed these in their work.

- Target Journal: Given the absence of the prodystroglycan complex, the study may not meet the high novelty bar of Nature Communications. It would be better positioned in a specialized journal focused on structural biology or enzymology.

The LARGE1-prodystroglycan interaction is transient ($K_D = \sim 1 \mu M$) and cannot be observed using single-particle cryo-EM.

11. ITC Data Inclusion

The manuscript includes data and methods for isothermal titration calorimetry (ITC), but the corresponding figure is missing. Including this figure would substantiate claims about the binding interactions and thermodynamics of substrate recognition. The absence of this figure detracts from the completeness of the study and should be addressed promptly.

We removed these data because the point of inflexion was not obvious given the low C-value for the interaction ($C = n[\text{titrand}]/K_D$; $\sim (8 \text{ to } 16 \mu M)/0.3 \mu M$, Supplementary Figure 15d). The titration precipitated (both at pH 6.6 and 7.4) when prodystroglycan was used as the titrant and LARGE1dTM₂ as the titrand – which would be the sensible orientation given that LARGE1 dimerizes. Precipitation upon titration is indicative of an interaction but prevents the extraction of quantitative data in ITC. However, the titration did not precipitate in the reverse orientation, but has a very shallow binding curve, with an indiscernible point of inflexion which also cannot yield the data necessary to extract stoichiometry or binding affinity. An example curve is shown below:

These data cannot be interpreted with confidence, so we are omitting them from the manuscript. We instead used microscale thermophoresis to extract the binding affinity, but could not extract stoichiometry (Supplementary Figure 15d).

Minor Issues and Recommendations:

1. Figure Numbering and Supplementary Data: Correct figure mislabeling (e.g., Figure 1c, 10-14 should be Figure 1c, 9-13 in the supplementary section) to ensure clarity and consistency.

The figure numbering has been corrected.

- Writing Style: Several sections of the manuscript are verbose and could be streamlined.

The writing style is intended to help readers who are not experts in the field by providing more information.

- Structural Comparisons: Adding comparisons to other glycosyltransferases, such as LARGE2 or related enzymes, would situate the findings within the broader context of glycosylation research.

We appreciate this suggestion and have included a section in the Discussion, where we compare our findings with published structures and length control mechanisms of other

glycosyltransferases (lines 288-295). The related enzymes have also been introduced in publications with references 36-40.

Reviewer #3

Remarks to the Author:

Reviewer #4

Remarks to the Author:

Matriglycan, a post-translational moiety on dystroglycan, plays important biological functions through binding with various ligands, for example, physical maintenance of skeletal muscle fibers, brain development, and Lassa fever virus infection. Mutations in proteins involved in matriglycan modification cause muscular dystrophy and lissencephaly. Although dystroglycan gene DAG1 and genes involved in matriglycan modification are ubiquitously expressed, matriglycan length (i.e., ligand binding activity) varies depending on the tissue and the cell type. However, it is unknown how matriglycan length is controlled. Elucidation of this question will lead to a deeper understanding of the mechanisms of glycosylation and muscular dystrophy. The authors propose a new mechanism of matriglycan modification through experiments in structural biology, biophysics, biochemistry, etc. However, there is insufficient data to draw the conclusion and proposal that the authors claim. Otherwise, the authors should weaken the claim or add more data. Some concerns are described below.

We thank reviewers three and four for the time they have taken to understand our work in the context of previous findings. You raise pertinent questions, and your suggestions for revision help strengthen the manuscript.

1. Throughout this article, the authors refer to the synthesis style of matriglycan as “processively” or “distributively”, but the meaning and definition of these terms should be clearly explained. I (general readers as well) may not have fully understood what the authors claim.

We agree with you and have added text in the introduction as a primer for the reader (lines 64-66) “dissociates from substrate between catalytic cycles”. We have also added the following sentence in the Results section to clarify our meaning (line 195 onwards): “Processive polymerization refers to the consecutive additions to a chain without dissociation between the substrate and the enzyme between catalytic cycles – these phenomena are typically observed in polynucleotide (DNA/RNA polymerases), polypeptide (ribosome) or polysaccharide syntheses”.

2. For in vitro matriglycan synthesis assay (Fig 4d, right gel), how did the authors make the DXD1:DXD3 heterodimer? Since the authors mentioned that there is no exchange between different homodimers in the solution, I understand that they did not mix separately purified Dx1 and DXD3. If they made it by co-transfection, the preparation would contain not only heterodimers but also homodimers (theoretically, DXD1 homo, heterodimer, and DXD3 homo are 1:2:1). In that case, wouldn't matriglycan synthesis proceed by the two types of homodimers contained in the preparation? This question is also for pseudomonomeric LARGE1dTM (Dead-Live heterodimer).

Thank you for pointing this out. We have added a line (#238-9) to explain our method more clearly. Briefly, we expressed FLAG-tagged DXD1 and His-tagged DXD3 translated with a P2A linker (which forces the ribosome to skip peptide bond formation) in HEK 293 Freestyle cells. We used tandem affinity purification of the protein to select for heterodimers of DXD1-DXD3. The same was done for the pseudomonomeric LARGE1 mutant.

3. In relation to the above comments, they said that there is no exchange because each homodimer of DXD1 and DXD3 is SDS-resistant, but this does not prove that exchange does not occur when DXD1 homodimer and DXD3 homodimer are mixed in solution. For example, why not mix homodimers tagged with different tags, such as DXD1-myc and DXD3-FLAG, and perform immunoprecipitation with one of the tags?

LARGE1dTM dimers can further oligomerize as revealed by Blue-Native PAGE. As such, protomer exchange will be confounded in co-IP assays with oligomers of dimers.

4. In the right gel of Fig. 4d, why does wildtype LARGE1dTM not produce a smear band of high molecular weight after a long-time reaction? As I mentioned before, this may depend on the definition of “processively” and “distributively”. If “processively” means extending matriglycan, then shouldn't wildtype LARGE1dTM produce a smear band that spreads to high molecular weight?

You correctly allude to the importance of the definition of distributivity/processivity (defined earlier). The nuance is that whether an enzyme remains associated with its substrate over consecutive catalytic cycles may be separate from the mechanism by which it controls length (degree of polymerization), but in this case appears to be intertwined.

Additional information: Most enzymes reside somewhere on a continuum between distributivity and processivity.

A formal definition of processivity relates to the probability of an enzyme dissociating from substrate at a given degree of polymerization. A processive enzyme may not extend polymers ad infinitum. Many processive polymerases rely on an orthogonal polymeric template to govern the length of the newly synthesized polymer; but other length-controlling mechanisms may be used in the case of template-free syntheses.

5. As shown in Fig. 5 and Supplementary Fig. 6, the T192M mutant is modified with matriglycan in HEK cells and mouse tissues, and its binding to LARGE1 and affinity of

UDP-GlcA to the ES complex are comparable to those of the wild type. However, it is very puzzling why *in vitro* polymerization does not occur at all (Supplementary Fig. 6b and c). The authors speculate that there is a different factor *in vivo*, but if so, the data is insufficient to claim that DGN regulates the length of matriglycan.

We agree with your observation and are also puzzled as to why in vitro polymerization does not occur. We changed the wording to “may aid” in line 266 since we don’t know for sure.

6. In addition, this contradicts the results of a previous report from the same group (New Engl J Med 2011) that the T192M mutation disrupts LARGE-dystroglycan interaction.

We appreciate the reviewer’s observation and agree that DAG1 T192M mutation interacts with LARGE1, as evidenced by the modification of this variant with matriglycan. The previous NEJM experiment appears to assume that wild-type and T192M constructs are expressed at the same level in cells and were normalized by applying total cell lysate for pulldown assays. Our findings indicate that the expression of T192M mutants is markedly lower than that of the wild-type protein, and concentrations need to be normalized (not to the cell lysate) for pulldown assays. So, although it may appear there is less binding of T192M, there is in fact, proportional binding in the previously published pull-down assay since there is a faint band in both mutant lanes. The newer assays comparing wild-type and T192M prodystroglycan binding to LARGE1 carry more weight since they are quantitative (Supplementary Figures 15d and e):

We compared the apparent migration of matriglycan synthesized endogenously by HEK 293 Freestyle cells on wild-type and T192M prodystroglycan constructs. Both constructs are exposed to the same factors in cells. Matriglycan polymerized on the T192M mutant construct migrates at a lower molecular weight.

7. The proposal that DGN functions as a molecular ruler is very interesting, but the only data leading to this proposal is dry data from Caver analysis and alphafold3 prediction. Even if DGN functions as a molecular ruler, it cannot explain the difference in matriglycan length between cell types and tissues in wild-type mice.

Dystroglycan carries O-glycans other than matriglycan, which are tissue-specific. The mucin-like domain of dystroglycan is modified by other O-linked sugars that may be tissue-specific, but the trend remains within each tissue type that T192M migrates at a lower MW than wild-type. To clarify the difference in matriglycan length, we added the phrase “heavily and heterogeneously O-glycosylated” in the introduction (line 46) and “heavily and heterogeneously O-glycosylated” in the figure legend (Figure 1b; lines 362-3).

8. It also cannot explain the phenomenon that the WT band is extended to a smear high molecular weight by overexpression of LARGE.

Overexpressing LARGE1 means prodystroglycan encounters LARGE1 spuriously many times and has the potential to lose matriglycan length control on removal of DGN by furin-like proteases, for which we have not removed the cryptic proteolytic site, such as disintegrins ADAM15 or ADAM17 (Figure 4b). This provides additional evidence that the DGN functions as a molecular ruler.

9. There are some parts where the description of the experimental method is insufficient and the results cannot be followed accurately. Specifically; Is the prodystroglycan a sample of 100mM DEAE elution; How was the LARGE1dTM-DAG1 reconstituted used for cryoEM and UDP titration assay?

We apologize for any confusion this may have caused. The prodystroglycan was a sample of 100mM DEAE elution only for matriglycan synthesis assays. Recombinant prodystroglycan was Ni IMAC (line 446-8) affinity-purified and then desalted into buffer using a PD10 column (added to lines 448-9 Methods section). No ion-exchange step was conducted; thus, a mixture of variously post-translationally modified products, LARGE1dTM and prodystroglycan was added together in a 2:1 ratio (line 522-3 Methods section).

10. In the schematic diagram of the glycan structure in Figure 1C, the glycan of prodystroglycan stops at xylose-glucuronic acid, just before matriglycan modification. This is probably true because matriglycan polymerization does not occur after b-GUS treatment as in the experiment in Figure 3, but there is no direct evidence that the schematic diagram is correct because there are no glycan analysis experiments.

You are correct. Figure 1c shows the subset of prodystroglycan that is receptive to LARGE1 modification, although the modifications may terminate at any point in the sequence. We have added the phrase “and is receptive to modification by LARGE1” to the figure legend to address the reviewer’s concern.

11. It would be better to include data showing that matriglycan modification does not occur in prodystroglycan expressed in B4GAT1-KO cells though it is also indirect evidence.

*We and others have previously published a manuscript describing the function of B4GAT1 as a primer for LARGE1 matriglycan synthesis (Willer *et al.* eLife 2014 and Praissman *et al.* eLife 2016) and added the references in line 177 of the manuscript.*

12. The following points need to be checked for accuracy and readability.
Lines 121 and 140, DAG1 28-340. Is 28-340 correct?

Yes, this was the construct used for cryo-EM analysis.

13. In Fig2e it says 28-749.

Yes, this was the structure used for MST analysis.

14. Lines 401 and 403, DAG1 28-340, 28-398. Are these correct?

DAG1₂₈₋₃₄₀ was changed to “constructs (as indicated)” (line 419).

15. The reference numbers in the text do not match the correct reference in the list.

The reference numbers have been corrected.

Point-by-Point Response to Reviewer Comments

We have amended our manuscript, including figures, according to the comments from the reviewers.

(Please see our responses in blue)

Reviewer #1

Remarks to the Author:

The authors have addressed all my comments.
I highly recommend accepting this revised manuscript for publication.

Thank you for re-reading the revised manuscript and providing encouraging comments.

Reviewer #2

Remarks to the Author:

Reviewer 2 – Second Round Report

I have read the corrected manuscript, the complete supplementary material, and the point-by-point rebuttal. The authors have invested substantial effort, and many of my earlier technical concerns are now addressed. The paper is clearer, and the new supplementary figures (especially the SAXS fits and AlphaFold confidence maps) are helpful. A few scientific and presentation issues still need attention before the work is ready for publication.

Overall Assessment

The study tackles a genuinely important question—how LARGE1 sets matriglycan length on dystroglycan—and combines cryo EM, SAXS, mutagenesis and enzymology rigorously. The conclusion that LARGE1 is processive on native prodystroglycan yet distributive on small substrates is well supported. The further suggestion that the DGN domain limits polymer length is plausible and interesting; however, the mechanistic link between DGN, enzyme dimer geometry, and chain length remains speculative.

The authors thank you for thoroughly re-reading the revised manuscript and for providing a detailed list of corrections, which are helpful to further improve the manuscript.

Major Comments

Comment Suggested Action

2.1 “Molecular ruler” language still too categorical. Figures 5f and Supplementary 18 rely on AlphaFold docking, and there is no direct density for DGN in the catalytic cleft. Re phrase throughout to make it explicit that the ruler model is hypothetical. E.g. change “DGN controls matriglycan length by acting as a ruler” → “our data are consistent with a ruler like role for DGN, although the structural details (Figs S18c are very small) remain to be elucidated.” It would also be helpful to label the respective XylT and GlcAT domains and catalytic site on the AF model of the complex (Fig S18d).

We have revised our language as suggested to indicate the ruler model is hypothetical by using the words, “potential function as a ruler” (line 279), “putative ruler,” (line 300) and have updated the phrasing in Figure

5, “acts as a ruler to measure length”, line 445. We have inserted this phrase, “Although the mechanistic details remain to be elucidated” into the discussion (lines 300-301) instead of the into the legend for Figure 5.

2.2 T192M: loss of O glycosylation vs. steric clash. Thr 192 could itself be O glycosylated; replacing it with Met might shorten matriglycan indirectly by removing that glycan. The rebuttal mentions this, but the manuscript does not. Add a short paragraph in the Discussion acknowledging this alternative explanation and, if possible, cite any (MS) data or prediction (eg. NetOGlyc) indicating whether or not Thr192 may carry an O glycan in WT dystroglycan.

There is potential for T192 to be O-glycosylated; however, there is currently no experimental evidence. Published work on O-glycosylation of dystroglycan (Stalnaker *et al.* 2010, Nilsson *et al.* 2010, Harrison *et al.* 2012 and Tran *et al.* 2012) only analyzes mature dystroglycan (residues 313 onwards), which does not include T192. The prodomain, DGN, is well-folded and likely inaccessible to glycosyltransferases. It has a low probability of glycosylation (0.0848265) using predictors such as NetOGlycan 4.0 whereas serine and threonine residues in the mucin-like domain average probabilities between (0.97-0.99) for O-glycosylation. We have added this information to the discussion (lines 303-306).

2.3 Connection between dimer asymmetry and length control is still vague. One xylosyl transferase domain is poorly resolved, but the functional implications are unclear. Add 1–2 sentences clarifying whether the authors think asymmetry itself gates processivity or is simply a cryo EM artefact, to aid non structural readers.

Our work does not test for a correlation between asymmetry in the dimer and length control, since we cannot generate a structurally symmetric dimer. However, there may be a connection between asymmetry and binding of DGN in a 2:1 LARGE1-prodystroglycan ES complex. We are unsure whether asymmetry directly gates processivity since we are unable to isolate a structurally symmetric LARGE1dTM dimer for such an experiment. To address this, we have added the text, “but we never observed nor could isolate the structurally symmetric dimer for comparison or to determine its function.” (lines 130-131).

Our data support that asymmetry of the LARGE1 dimer is biologically relevant and not a cryo-EM artefact. Our data further suggest that entropy/disorder is random exchange – it doesn’t necessarily matter which protomer is the “entropic counterweight” as long as one of them carries the burden of organizational energy for the other.

2.4 LARGE1 vs. LARGE2 context. A sequence alignment is now provided (Supp. Fig. 4), but functional differences between the paralogues are not discussed. Insert 2–3 sentences in the Introduction or Discussion summarising known LARGE2 activity and why it was not pursued here.

LARGE2 was not pursued further in this study because, although both proteins are capable of synthesizing matriglycan, primarily LARGE1, but not LARGE2, is expressed in excitable tissues. To clarify this, we have inserted the following sentence in the introduction, “LARGE2dTM was not analyzed further in this study because deletion of LARGE1, but not LARGE2, causes neuromuscular pathology” (lines 142-143).

2.5 Proposed N glycan structure lacks the conserved trimannose core. A dimannose core would be highly atypical for a complex glycan. Provide experimental evidence (e.g., MS/MS, NMR, targeted exoglycosidase digestion) to support the assignment; clarify whether the glycan occurs naturally or is synthetic; outline a plausible biosynthetic route; and confirm that the omission of the two mannose residues is not a typographical error.

The MS annotation was based on observed MS/MS fragments (see Supplementary Fig. 5) and common glycan structures found in HEK cells, but it is ambiguous. We have stated the ambiguity in the figure legend.

The conserved N-glycan core was annotated for all MS data. The three green circles represent the mannose residues, so all N-glycan cores that we present share the conserved structure. The bisecting GlcNAc might be confusing as MGAT3 expression in HEK cells is very low. We have indicated with a red background region that the additional GlcNAc could locate at one of the antennae.

2.6 UDP GlcA only MST assay: the primer ends in GlcA, so UDP GlcA could bind non productively; absence of a UDP Xyl control leaves the 30 fold affinity increase open to alternative interpretation. Supply MST data (or a rationale) for: (i) **pre incubation with UDP Xyl to generate a xylose terminated acceptor**, and/or (ii) simultaneous UDP Xyl + UDP GlcA conditions; explain how these controls support the current Kd interpretation.

We calculated the K_D of UDP-GlcA for the ES complex and showed that it does not bind LARGE1 in solution. Our completed experiment, as suggested, (i) **pre incubation with UDP Xyl to generate a xylose terminated acceptor (Reviewer-only Figure 1a, below)**. We show that there is no significant difference in binding affinity to UDP-GlcA when matriglycan terminates in xylose or glucuronic acid. We additionally show that the affinity of the LARGE1-prodystroglycan ES complex for UDP-GlcA remains the same whether we used the prodomain expressed in HEK 293 Freestyle cells (DAG1₂₈₋₇₄₉) modified by matriglycan or DGN expressed in bacteria (DAG1₅₀₋₃₁₃; Bozic 2004), which terminates prior to the mucin-like domain and lacks matriglycan (Reviewer-only Figure 1b, below). This suggests that the only factor governing the binding affinity of UDP-GlcA to LARGE1 is the presence of the prodomain and not matriglycan, nor its terminating sugar. We also show that prodystroglycan (DAG1₂₈₋₇₄₉) binds to LARGE1dTM with the same affinity in the presence or absence of UDP-xylose but increases its affinity for prodystroglycan in the presence of UDP-GlcA (Reviewer-only Figure 1c, below).

Reviewer-only Figure 1. UDP-glucuronate binds LARGE1dTM-prodystroglycan complexes regardless of the presence of matriglycan or terminating sugar. **a.** Prodystryglycan (DAG1₂₈₋₃₄₀) was complexed with NHS-red-labelled LARGE1dTM and preincubated with UDP-xylose (DAG1₂₈₋₃₄₀-Xyl-L1dTM) or no additive (DAG1₂₈₋₃₄₀-L1dTM) for one hour and was subsequently titrated with UDP-glucuronate (UDP-GlcA). No difference in binding affinity was observed. **b.** MST titration of NHS-red-labelled LARGE1dTM complexed to prodystroglycan constructs either expressed in mammalian cells and post-translationally modified by matriglycan (DAG1₂₈₋₇₄₉), or *expressed in bacteria* (DGN, DAG1₅₀₋₃₁₁) lacking the mucin-like domain, and therefore matriglycan, in the presence of UDP-glucuronate. **c.** MST titration of NHS-red-labelled LARGE1dTM with prodystroglycan (DAG1₂₈₋₇₄₉) in the presence of UDP-sugars, as indicated.

As a caveat, we cannot assume that all recombinant prodystroglycan expressed in HEK 293 Freestyle cells terminates in xylose-glucuronate. We have illustrated our schematic to show the Xyl-GlcA primer as the substrate in enzymatic assays because it is the only species reactive to LARGE1. However, the recombinant prodystroglycan used for MST is heterogeneously glycosylated as it was purified using nickel affinity followed by desalting and we did not select for fractions without matriglycan. Therefore, recombinant prodystroglycan can theoretically have any degree of glycosylation, ranging from no core mannose 1-3 at any of the relevant positions to complete glycosylation including endogenously synthesized full-length matriglycan.

The ITC data (about 1,2 uM Kd) on Large1dTM-DAG1 titration with GlcA may indicate a displacement event (this would be in line with a processive transfer action) that might be fitted using a competitive binding model using more advanced software (eg. AFFINIMETER)?

The ITC data is too ambiguous to draw definitive conclusions given that there is no obvious point start/end plateau nor a definitive point of inflection. Given this limitation, even using a competitive binding model with more advanced software would not result in unambiguous conclusions.

2.7 Supplementary Fig. 4 (line 122 p. 6) is Fig S3 in the supplementary materials and only shows part of the GlcA monosaccharide part. Correct and remediate by showing the whole UDP-GlcA donor sugar, similar to the zoom-in in Fig 2d.

We appreciate the reviewer pointing out this error and have made the correction in the text (line 122). We have included another view from a different angle to show density for glucuronate (Supplementary Fig. 3b).

Minor Comments:

1. Figure 1a – The Poisson fit is now in Supp. Fig. 1; please reference it explicitly in the legend.

We have included a reference to Supplementary Fig. 1 in the legend for Fig. 1 (line 364).

The DAG1 construct used to make the complex with Large1dTM needs to be called like that in the Figure (1b): is DAG1 the second structure (Golgi, with DGN) or third structure (cytosolic, upon cleavage by furin)?

We have added clarification for the prodystroglycan construct used in Fig. 1c by adding “DAG1₂₈₋₇₄₉” to the legend of Fig. 1 (line 379) and we have added the construct used for the cryo-EM reconstruction of LARGE1-prodystroglycan by including “DAG1₂₈₋₃₄₀” in the legend for Fig. 2 (line 401). For an explanation: DAG1 is name of the gene that encodes dystroglycan. Figure 1b shows the post-translational modifications that the DAG1 gene product undergoes as it passes through the secretory system (ER and Golgi), including glycosylation and autoproteolytic cleavage into alpha and beta chains, to produce mature dystroglycan that lacks the prodomain (DGN).

We see our mistake in calling the construct in Figure 1b “prodystroglycan” and have changed it to “dystroglycan” line 370. As the prodomain (DGN) is necessary to obtain the relevant post-translational modification, any construct that contains the prodomain (DGN; residues 28-312 encoded by DAG1) might be called “prodystroglycan” but may also be referred to as the following: DAG1₂₈₋₇₄₉, DAG1₂₈₋₃₉₈ or DAG1₂₈₋₃₄₀. We have used DAG1 with subscript numbers to make explicit the domain boundaries of constructs used in experiments in the legends for Figures 1 (line 379) and 2 (line 401).

2. Figure 3 legend – good addition of sugar symbols, but please define “Xyl GlcA” directly in the caption for first-time readers.

Thank you for pointing this out. We have added “xylose-glucuronate (Xyl- β 1,4-GlcA)” in line 162 to clarify the abbreviation for first-time readers.

3. Supplementary Fig. 11 – include χ^2 values in the panel or caption to make goodness of fit immediately visible.

The χ^2 values for each CRY SOL and FoXS fitting are individually inset into each panel.

4. Supplementary Fig. 16 – Explain what is C1 and C2 symmetry: simply introducing a 2-fold or more (like C-centering?). Add PDB codes of obtained structures in the legend.

We have drawn a dotted line indicating the axis of two-fold rotational symmetry on the reconstructed volume within the workflow and added the sentence, “The axis of two-fold rotational symmetry (C2) is by the dotted line on the reconstructed volume on the right-hand-side”, to the legend for Supplementary Fig. 16. We have also added the PDB IDs for volumes reconstructed in C1 and C2 symmetries, 6UI7 and 7UI7, respectively, to the figure and legend.

5. Supplementary Fig. 17 – Large1 is bound to more than UDP along, or can this not be inferred? Please add the PDB codes for the obtained structures.

There is sufficient density to model glucuronate. We have updated the label in Supplementary Fig. 17 to reflect this.

6. Typos

1. line 302, “biding” should be “binding”. corrected
2. Line 239 active sites on one protomer is -> are 239 mutated. corrected
3. Line 309 “arena viral infection” should be “arenaviral infection”. corrected
4. Caption Fig 5f – “Threonine 192 is show in red” → “is shown”. corrected
5. Line 453 - “SEC MALS SAX was performed” should be “SEC MALS SAXS”. corrected

Reviewer #3

Remarks to the Author:

Reviewer #4

Remarks to the Author:

My concerns have been appropriately addressed in this revision. It would be very helpful for readers to include the following points; 1) methods to obtain heterodimers of DXD1-DXD3 as the authors explained in response #2, and 2) a possible explanation why wt LARGE1dTM did not produce a smear band of high mw after longer incubation as the authors explained in response #4, i.e., “A processive enzyme may not extend polymers ad infinitum. Many processive polymerases rely on an orthogonal polymeric template to govern the

length of the newly synthesized polymer, but other length-controlling mechanisms may be used in the case of template-free syntheses”.

Thank you for stating that we have answered your questions satisfactorily. The thoughtfulness of your questions was helpful for our future work.